


# The depth scales of the AMOC on a decadal timescale

Tim Rohrschneider[1,2], Johanna Baehr[3], Veit Lüschow[1], Dian Putrasahan[1], and Jochem Marotzke[1,4]

[1]Max Planck Institute for Meteorology, Hamburg, Germany
[2]International Max Planck Research School on Earth System Modelling, Hamburg, Germany
[3]Institute of Oceanography, Center for Earth System Research and Sustainability, Universität Hamburg, Hamburg, Germany
[4]Center for Earth System Research and Sustainability, Universität Hamburg, Hamburg, Germany

**Correspondence:** Tim Rohrschneider (tim.rohrschneider@mpimet.mpg.de)

**Abstract.** We use wind sensitivity experiments to understand the wind forcing dependencies of the level of no motion and the e-folding pycnocline scale as well as their relationship to northward transport of the mid-depth Atlantic meridional overturning circulation (AMOC) south and north of the equator. In contrast to previous studies, we investigate the interplay of nonlocal and local wind effects on a decadal timescale. We use 30-year simulations with a high-resolution ocean general circulation model

5 (OGCM) which is an eddy-resolving version of the Max Planck Institute Ocean Model (MPIOM). Our findings deviate from the common perspective that the AMOC is a nonlocal phenomenon only, because northward transport in the inter-hemispheric cell can only be understood by analyzing nonlocal Southern Ocean wind effects and local wind effects in the northern hemisphere downwelling region where Ekman pumping takes place. Southern Ocean wind forcing predominantly determines the magnitude of the pycnocline scale throughout the basin, whereas northern hemisphere winds additionally influence the level of no motion

10 locally. In that respect, the level of no motion is a better proxy for northward transport and mid-depth velocity profiles despite the Ekman return flow which is found to be baroclinic. We compare our results inferred from the wind experiments and a 100-year global warming experiment in which the atmospheric $CO_2$ concentration is quadrupled, using MPIOM coupled to an atmospheric model. We find that the evolution of the level of no motion in response to global warming represents changes in vertical velocity profiles or northward transport, whereas the changes of the pycnocline scale are opposite to the changes of the

15 level of no motion over time. Using the level of no motion as depth scale, the analysis of the wind experiments and the warming experiment suggests a hemisphere-dependent scaling of the strength of AMOC. Furthermore, we put forward the idea that the ability of numerical models to capture the spatial and temporal variations of the level of no motion is crucial to reproduce the mid-depth cell in an appropriate way.





## 1 Introduction

To date and despite a wide range of theoretical and experimental studies, we do not fully understand inter-hemispheric over-turning of the mid-depth cell in the Atlantic and the role of southern and northern hemisphere processes like the response to changes in the surface winds. Current understanding of the Atlantic meridional overturning circulation (AMOC) suggests an interplay between adiabatic pole-to-pole overturning (e.g. Toggweiler and Samuels, 1995; Wolfe and Cessi, 2011) and low-latitude diabatic forcing that establishes a balance between downwelling of heat and upwelling of deep waters (e.g. Munk and Wunsch, 1998; Marotzke, 1997). Especially in connection to Southern Ocean processes and the Antarctic circumpolar current (ACC), the effect of winds on the AMOC and basin-wide density stratification has gained considerable attention during the last two decades (e.g. Marshall and Speer, 2012; Johnson et al., 2019). Poorly understood, however, is the influence of winds on inter-hemispheric overturning in the Atlantic away from the surface Ekman layer. This paper presents an analysis of wind sensitivity experiments in order to provide insight into the inter-hemispheric circulation by understanding the behavior of the depth scale(s) of the AMOC. We focus on the response of the AMOC to wind forcing on a decadal timescale after the realization of major adjustments. To further enhance our conceptual understanding, we compare the outcome of these wind sensitivity experiments and the outcome of a global warming experiment.

Oceanographers use theoretical scaling relationships to provide conceptual understanding and to estimate the strength of the AMOC in response to different forcings. Nowadays, the most common analytical model to describe meridional overturning of the upper branch of the AMOC is the pycnocline model (Gnanadesikan, 1999). According to this model, the vertically integrated northward transport in the northern hemisphere is proportional to the basin-averaged e-folding pycnocline scale. The pycnocline scale is a measure for density stratification and describes how density unfolds vertically. The deeper the pyc-nocline scale, the stronger the transport, with the assumption that zonal or meridional density gradients are fixed. The depth scale itself is determined by Southern Ocean winds and eddies, diapycnal upwelling in the tropics, and North Atlantic deep water formation. In the same manner, a wide range of theoretical studies or scaling arguments rely on the assumption that the pycnocline scale translates a zonal or meridional density gradient into horizontal force that drives northward flow (e.g. Robin-son and Stommel, 1959; Bryan, 1987; Marotzke, 1997; Marotzke and Klinger, 2000). The majority of the different scalings makes use of the thermal wind relation which is based on the geostrophic and hydrostatic approximations of the momentum equations. The thermal wind relation is also key for reconstructing the AMOC strength using observations which are boundary densities (Hirschi et al., 2003; Baehr et al., 2004; Hirschi and Marotzke, 2007; Baehr et al., 2009). In general, the theoreti-cal and experimental studies have in common that they integrate zonal or meridional density gradients twice in the vertical, $\psi \propto \Delta\rho\, \eta^2$, where $\psi$ is the strength of northward overturning, $\Delta\rho$ is the density gradient, and $\eta$ is the depth scale. Differ-ent assumptions like advective-diffusive balance modify the simple scaling relationship considered here. Depending on the timescale, various studies further relate the strength of the AMOC to density gradients and depth scales in specific regions (e.g. Schloesser et al., 2012, 2014; Butler et al., 2016). The nonequivalance of different depth scales of the AMOC has first been noted in Scott (2000), and recent studies put into question whether the pycnocline scale is the appropriate depth scale to





estimate the AMOC strength. For instance, Levermann and Fuerst (2010) find that the pycnocline scale and meridional density
gradients are mutually independent. Using a coarse-resolution model, Griesel and Maqueda (2006) show that the pycnocline
scale does not scale northward transport in experiments in which density gradients are artificially modified. In this connection,
DeBoer et al. (2010) find that the depth of maximum overturning (level of no motion) is a more appropriate parameter to scale
maximum overturning. Shakespeare and Hogg (2012) use the depth scales of the overturning extrema to build an analytical
model for both the mid-depth cell and the abyssal ocean. Finally, Marshall and Johnson (2017) combine the depth of maxi-
mum overturning (level of no motion) and the e-folding pycnocline scale to express the relative strengh of the ACC and the
AMOC. The present study adresses the wind forcing dependencies of the level of no motion, pycnocline scale and northward
transport in the inter-hemispheric region as well as the response of these quantities to global warming. We analyze these de-
pendencies on a decadal timescale to explore experimentally to which degree the depth scales represent northward transport of
the inter-hemispheric cell. In this connection, the study is based on different ways or definitions which describe meridional flow.


Theory of wind driven changes in stratification can be traced back to the development of the planetary geostrophic equations
(e.g. Welander, 1959; Bryan, 1987) and the related theory on the ventilated thermocline in the subtropical region (Luyten et al.,
1983). In a comprehensive paper on the production of stratification, Vallis (2000) brings together the scaling of the advective
depth scale of the subtropical thermocline due to Ekman pumping and the scaling of the diffusive layer below the advective
depth. Scaling of the diffusive thermocline depends sensitively on the Ekman pumping velocity and the advective depth scale,
in the sense that the diffusive layer is related to the scale of sloping isopycnals. Common scaling suggests that the level of no
motion is far below the advective depth. However, during the course of this study we show how wind forcing influences the
level of no motion and deep velocity profiles locally. Changes in stratification that have their origin in changes in the surface
wind stress are not limited to lower latitudes. The absence of continental barriers in the Southern Ocean and the strong input
of momentum at the surface establishes a deep reaching Ekman overturning cell. The steepening of isopycnals, which is com-
pensated by baroclinic instability that induces an eddy field, is thought to influence deep stratification and northward transport
throughout the basin (e.g. Vallis, 2000; Klinger and Cruz, 2009; Allison et al., 2011). The zonal peridiocity of the ACC boosts
the strength of the AMOC in response to an increase in Southern Ocean wind forcing (Klinger et al., 2003, 2004). However,
the local influence of northern hemisphere winds on the AMOC is less understood. Tsujino and Suginohara (1998) propose
an enhancement of the thermohaline circulation due to a wind-driven buoyancy gain in the upwelling region of the northern
hemisphere. Recently, Cessi (2018) finds that the inter-hemispheric cell weakens in response to increased westerlies at the
northern high latitudes. The local influence of wind forcing in the northern hemisphere on the AMOC is commonly ignored in
the theoretical literature on the AMOC. A study by Cabanes et al. (2008) already indicates that wind stress curl variations may
play a crucial role in setting the AMOC shear component. In that respect, we focus on the effects of the winds over the North
Atlantic downwelling region which emerges from Ekman pumping at lower and mid-latitudes. We address the question how
changes in both nonlocal and local wind forcing influence the depth scales and northward transport in the inter-hemispheric
region. We also point out how the relationships between these variables change in response to global warming. We expect that
the level of no motion shoals and the strength of the mid-depth cell weakens in the warming experiment on a multi-decadal



timescale (e.g. Zhu et al., 2015; Jansen et al., 2018). It is uncertain, however, to which degree decadal changes in vertical
velocity profiles or shear are related to changes in the level of no motion and the pycnocline scale.

To answer the specific questions that are outlined above, we use 30-year wind experiments with a high-resolution ocean general circulation model (OGCM) and a 100-year global warming experiment in which the OGCM is coupled to an atmospheric model. We expect a robust response of the AMOC on a decadal timescale because major adjustments by basin-wide wave
propagation are realized on this timescale. To the best of our knowledge and despite a wide range of theoretical and modeling studies on the Atlantic circulation, the specific questions have never been answered in an explicit way. That is to say, from an overarching perspective we analyze whether the depth scales are proxies for northward transport in the inter-hemisperic cell, and we ask whether we should adapt a more nonlocal or local perspective with respect to inter-hemispheric overturning and hemispheric differences.


In the following section we briefly describe the different experiments and the experimental strategy. In section 3 we describe the differences in density stratification with the changes in wind forcing as well as the wind forcing dependencies of the level of no motion, the pycnocline scale and northward transport. In section 4 we analyze the relationship between velocity profiles or shear and the depth scales from a local and a more nonlocal perspective on hemispheric differences. Finally, we compare
the outcome of the wind experiments and the outcome of a warming experiment, in section 5.

## 2   Experiments and methods

### 2.1   Numerical model and experiments

We use wind experiments conducted with a vertically and horizontally high-resolution, eddying configuration of the Max Planck Institute for Meteorology ocean model (MPIOM). The version is called TP6ML80 and has been developed within the
German Consortium project STORM (von Storch et al., 2012). It is based on a tripolar grid with a horizontal resolution of 0.1 degrees and a vertical resolution of 80 unevenly spaced levels. Compared to low-resolution MPIOM versions, we assume better model physics in terms of the resolution of mesoscale eddies as well as wave propagation.

We have available three different wind experiments with realistic geometry (Lüschow et al., 2020). In the 1X experiment,
the standard surface momentum and buoyancy fluxes from NCEP-NCAR reanalysis-1 (Kalnay et al., 2018) are applied. In the 2XSH experiment, the monthly climatology of the zonal and meridional surface wind stress is doubled only over the Southern Ocean. By applying a sine function which declines to zero towards the equator in the southern hemisphere, we obtain a smooth transition of the surface wind stress towards the original surface momentum fluxes. In the 2X experiment, we double the monthly climatology of the zonal and meridional surface wind stress in both the northern and southern hemispheres throughout
the basin. Fig. 1 shows the climatology of the zonal surface wind stress of the different experiments. The wind experiments are initialized with the state of the 1X experiment, and the wind forcing is switched on at year 1980. Because the computational




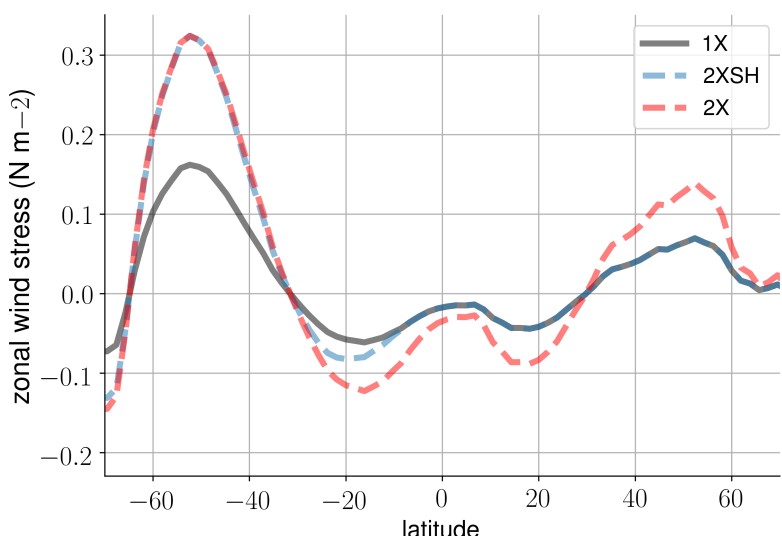

**Figure 1.** The zonal-mean zonal wind stress in the 1X experiment (black), 2XSH experiment (blue), and 2X experiment (red) (Lüschow et al., 2020).

costs for an eddy-resolving simulation are still large, the simulation period is only until year 2010. Using monthly mean output, we focus on the time-window 1991 to year 2010 after the realization of major adjustments to the changes in the surface wind stress in order to analyze the AMOC response to forcing.


Like the different wind experiments, the warming experiment (4XCO2) is conducted with a horizontally high-resolution configuration of MPIOM (Putrasahan et al., 2020). The version is called TP6ML40 and it is coupled to the atmospheric model ECHAM6. The coupled model is described in Gutjahr et al. (2019) and initialized with the state of the control simulation which mimics observed conditions at year 1950. In constrast to TP6ML80, the ocean model has only 40 unevenly spaced lev-

els because we have a 100-year simulation and ease the computational burden. We expect that the level of no motion changes more strongly in response to global warming than in the wind experiments and that the details of the vertical model grid do not influence the conceptual findings on the warming experiment. In order to have a large signal, a quadrupling of the 1950 $CO_2$ concentration is used. After a 30-year spin-up, the coupled model is integrated from year 1980 to year 2079.

## 2.2 Experimental strategy and quantities

The AMOC is described by the overturning streamfunction in the latitude-depth section, $\psi(t,z,y) = \int_{z=0}^{z=D} \int_{x_e}^{x_w} v(t,z,y,x)\,dx\,dz$, where $v$ is the meridional velocity, $z$ is the depth with $D$ the depth at the bottom of the ocean column, and $x_e$ and $x_w$ are the



**Figure 2.** On the left, we show the time-mean AMOC streamfunctions (1991-2010) in the (a) 1X experiment, (b) 2XSH experiment, and (c) 2X experiment. On the right, we show the evolution of the AMOC in the 4XCO2 experiment using (d) years 1-20, (e) years 41-60, and (f) years 81-100.





eastern and western boundaries of the Atlantic basin. The question arises why we rely specifically on the different experiments on a decadal timescale, and in this section we elaborate on the experimental strategy. We focus on the upper, northward

flowing branch of the mid-depth cell which rotates clockwise. Fig. 2a,b,c show the time-mean (1991-2010) overturning cells in the wind experiments. We analyze the AMOC in the inter-hemispheric region south and north of the equator. In this way, we explore the nonlocal response to changes in Southern Ocean winds and local wind effects in the downwelling region of the northern hemisphere. In general, the mid-depth cell strengthens with higher wind forcing over the Southern Ocean, but we cannot capture the details of the different experiments in terms of spatial variations. The surface meridional Ekman flux

can be inferred from the surface levels of the overturning streamfunction; it scales with the zonal wind stress and is inversely proportional to the Coriolis parameter. In contrast to the surface Ekman flux which is negative south of the equator and positive north of the equator, the northward flow of the mid-depth cell seems to be continuous and contiguous throughout the basin, and it is difficult to base inferences on purely regional dynamics. However, these surface Ekman fluxes already indicate that the flow is not as continuous as the AMOC streamfunction suggests, because they have to be compensated by an interior return

flux which changes the force balance of the flow. Furthermore, the wind stress curl over the basin imposes a forcing that may change stratification locally, in the sense that the meridional transport and its depth differ between the wind experiments. In the 4XCO2 experiment, the surface buoyancy fluxes change continuously. On a multi-decadal timescale, the mid-depth cell weakens and shoals after the forcing is switched on (Fig. 2d,e,f). The 100-year simulation time series makes it possible to analyze multi-decadal changes and compare the 4XCO2 experiment and the wind experiments. The warming experiment provides

conceptual understanding of the linkage between ocean heat uptake and changes in the depth scales of the AMOC and how they relate to northward transport. We summarize the experimental strategy as outlined above in Table 1.

We limit our analysis of the different experiments to a set of quantities (Table 2). The level of no motion of the mid-depth cell is the depth of maximum overturning where zonally averaged velocities reverse in sign, $\eta_\psi = z_{\psi(\max)}$. We use the monthly

mean outcome of the related quantities and subsequently take the time-mean (1991-2010). In doing so, we describe in more detail the behavior of the level of no motion and its relationship to northward transport. The reader should notice that the level of no motion incorporates nonlinearities because it is an integrated quantity and related to the zonal-mean meridional velocity. The level of no motion is not a zonal-mean quantity that is averaged over the zonal extent. Further, we avoid that the algorithm selects a model level within the depth range of the surface Ekman layer or perturbations that emerge from equatorial

upwelling. The pycnocline scale is the single mode e-folding scale for vertical density stratification, and the profile of the latter is assumed to be exponential and self-similar. We use the following fitting algorithm at each grid cell and subsequently take the zonal and temporal mean, $\eta_\rho = 2 \frac{\int_{z_r}^0 (\rho - \rho_r)\, z\, dz}{\int_{z_r}^0 (\rho - \rho_r)\, dz}$ (Gnanadesikan, 1999; Gnanadesikan et al., 2007; DeBoer et al., 2010). The reference depth is $z_r = -2500$ meters, and $\rho_r$ is the potential density at the reference depth. Using this algorithm we take the e-folding scale twice. In this way, 80 to 90 percent of the vertical density change is scaled, which scales the upper branch of the

mid-depth cell. In some regions, especially at the lateral margins of the basin, the profile for vertical density stratification may become more linear and deviate from an exponential profile. However, such deviations from a perfectly exponential profile do not restrict the ability of the pycnocline scale to interpret density stratification in different regions in the sense that it is an inde-



**Table 1.** Overview of the experimental strategy

| | |
|---|---|
| 1X (1991-2010) | The reference experiment which mimics observed conditions |
| 2XSH (1991-2010) | We double the monthly climatology of the zonal and meridional surface wind stress over the Southern Ocean only in order to analyze the nonlocal effects of Southern Ocean winds |
| 2X (1991-2010) | We double the monthly climatology of the zonal and meridional surface wind stress throughout the basin in order to analyze the interplay between nonlocal and local wind effects and extract local wind forcing dependencies |
| 4XCO2 (1980-2079) | We quadruple the atmospheric $CO_2$ concentration of the year 1950 in order to compare wind-forced and buoyancy-forced changes |

**Table 2.** Experimental quantities

| | |
|---|---|
| $\psi$ | AMOC streamfunction |
| $\rho$ | Potential density |
| $\eta_\psi$ | Level of no motion |
| $\eta_\rho$ | Pycnocline scale |
| $\eta_\mathrm{w}$ | Advective depth scale |
| $\psi_\mathrm{t}$ | Total maximum overturning streamfunction |
| $\psi_\mathrm{g}$ | Geostrophic approximation of the maximum overturning streamfunction |
| $\frac{\partial \psi}{\partial z}$ | Vertical derivative of the overturning streamfunction (velocities) |
| $\frac{\partial^2 \psi}{\partial z^2}$ | Second vertical derivative of the overturning streamfunction (shear) |





pendent measure as long as density increases with depth. We make use of a third depth scale and compute the advective depth $\eta_\mathrm{w}$ in order to provide the linkages between the differences in the wind stress curl, the differences in the density field $\rho$, and

the depth scales. Classical scaling of the thermocline equations suggests that the advective depth scale is directly proportional to the square-root of the local Ekman pumping velocity $W_\mathrm{E}$ (e.g. Vallis, 2000), $(\frac{W_\mathrm{E} f L^2}{g'})^{0.5}$, where $f$ is the Coriolis parameter, $L$ is the basin width, and $g'$ is the reduced gravity. We compute the local Ekman pumping velocity $W_\mathrm{E}$ from the wind stress data of the wind experiments.

During the course of our study we analyze both the total maximum overturning streamfunction $\psi_\mathrm{t}$ and the geostrophic approximation of the maximum overturning streamfunction $\psi_\mathrm{g}$; that is, the total streamfunction minus the surface Ekman flux. We compute both $\psi_\mathrm{t}$ and $\psi_\mathrm{g}$ because their wind forcing dependencies should be fundamentally different due to the interior, geostrophic return flow of the surface Ekman flux. In this way, we analyze the relationship between overturning and its depth from the perspective on integrated transport. We use the zonal surface wind stress $\tau_\mathrm{x}$ to compute the strength of the

surface meridional Ekman transport, $\int_{x_e}^{x_w} -\frac{\tau_\mathrm{x}(x,y,t)}{\rho_0 f} dx$, where $\rho_0$ is the reference density and $f$ the Coriolis parameter. We do not substract the interior return flow of the surface Ekman flux. For the sake of simplicity a range of studies assumes that the interior return flow of the surface Ekman flux is barotropic even on longer than a monthly timescale (e.g. Hirschi and Marotzke, 2007; Moreno-Chamarro et al., 2016). Based on an idealized experiment with an OGCM, a recent study demonstrates that the return flow is baroclinic and has strong contributions at the upper levels of the ocean on the timescale considered here (Williams

and Roussenov, 2014). On a monthly timescale, anomalies of the return flow are barotropic because the density field does not adjust (e.g. Jayne and Marotzke, 2001), but these anomalies are negligible in the set-up presented in this study and by their nature do not change the time-mean outcome. Later on, we support the perspective that the interior return flow of the surface Ekman flux is baroclinic. Finally, we analyze whether there are causal linkages between the changes in the depth scales and northward transport at different depths. Therefore, we compute the vertical derivative of the AMOC streamfunction $\frac{\partial \psi}{\partial z}$, which

represents the zonal-mean meridional velocity scaled by the basin-width. We then compute the second derivative of the AMOC streamfunction $\frac{\partial^2 \psi}{\partial z^2}$, which represents the vertical velocity shear scaled by the basin-width. In this way, we analyze to which degree the changes in the northward transport of the mid-depth cell are directly related to the displacement of the level of no motion. The computation of $\frac{\partial \psi}{\partial z}$ and $\frac{\partial^2 \psi}{\partial z^2}$ below the surface Ekman layer is independent of the way how we approximate the maximum overturning streamfunctions $\psi_\mathrm{t}$ and $\psi_\mathrm{g}$.

## 200 3 Wind forcing dependencies

### 3.1 Wind-driven changes in stratification

As a starting point, we highlight the differences in density stratification in the Atlantic basin between the wind experiments. Afterwards, we relate these differences to the level of no motion $\eta_\psi$ and the pycnocline scale $\eta_\rho$. We normalize the zonal-mean potential density to highlight the differences in density stratification between the wind experiments, $\frac{\rho_0-\rho}{\rho_0}$, with the reference

density $\rho_0 = 1025$ kg m$^{-3}$. Fig. 3a,b show the time-mean difference in density stratification between the 2XSH and 1X ex-





**Figure 3.** Differences in the zonal-mean density stratification averaged over the years 1991-2010: (a) the difference between the 2XSH and 1X experiments and (b) the difference between the 2X and 1X experiment. The black contour lines represent the zonal-mean density stratification $(10^{-1})$ in the 1X reference experiment. Density stratification is computed by $\frac{\rho_0 - \rho}{\rho_0}$, with the reference density $\rho_0 = 1025$ kg m$^{-3}$. In (c) we show the advective depth scale $\eta_w$ in meters depth in the 1X experiment (black) and the 2X experiment (red), with the reduced gravity set to $g' = 0.013$ (opaque) and $g' = 0.02$ m s$^{-2}$ (transparent). The smaller the value for the reduced gravity, the deeper the advective depth scale $\eta_w$.





periment and the 2X and 1X experiment. Fig. 3c illustrates the advective depth scale $\eta_\mathrm{w}$ in the Atlantic. Within the range of the advective depth scale $\eta_\mathrm{w}$, isopycnals shoal towards the equator due to equatorial divergence and deepen in the subtropical region towards higher latitudes due to the local forcing that is imposed by the wind stress curl. Deep stratification below $\eta_\mathrm{w}$ but within the depth range of the upper, northward flowing branch of the mid-depth cell reveals the same behavior in the southern
hemisphere, but isopycnals rise constantly towards the region of North Atlantic deep water formation.

The time-mean differences in density stratification between the wind experiments mirror the experimental set-up of the present study. In general, Fig. 3 suggests that the difference in density stratification between the 2XSH experiment and the 1X experiment is driven by the wind stress curl over the Southern Ocean, which has a strong influence in the southern hemisphere,
but deep isopycnals change even in the northern hemisphere. The change of the 2X experiment relative to the 1X experiment is driven by the change of the climatology of the wind stress curl in both hemispheres. Local Ekman pumping displaces isopycnals downward north of the equator with a maximum change in the subtropical region, and this displacement is scaled by $\eta_\mathrm{w}$. The influence of the local wind forcing prevails but deep isopycnals are also influenced nonlocally below $\eta_\mathrm{w}$. Both nonlocal and local wind effects change the density field, and their relative influence on stratification depends on location and
depth. We expect considerable changes in the level of no motion $\eta_\psi$ with changes in wind forcing in both hemispheres, and the mechanism how the changes in density stratification translate into changes in the level of no motion $\eta_\psi$ is simple from a generic point of view. Isopycnals are displaced downward by the wind forcing over the Southern Ocean, but the displacement of the isopycnals below $\eta_\mathrm{w}$ is small in the sense that the change in deep vertical velocity shear is presumably small. In the northern hemisphere, we find strong differences in density at the advective depth, which suggest that the velocities at mid-
depth are altered. Together with roughly constant vertical velocity shear at deeper levels, even small changes in the zonal pressure gradients at mid-depth should be related to substantial differences in the level of no motion $\eta_\psi$. In the following, we analyze how these changes in density stratification translate into changes in both depth scales $\eta_\psi$ and $\eta_\rho$.

### 3.2 Level of no motion and pycnocline scale

Fig. 4 shows the meridional dependence of the level of no motion $\eta_\psi$ and the zonal-mean pycnocline scale $\eta_\rho$ in the wind
experiments in order to explore the wind forcing dependencies of the depth scales. Considering the 1X experiment, the spatial variations of $\eta_\psi$ and $\eta_\rho$ coincide in the sense that both shoal towards the equator in the southern hemisphere and stay more or less constant or change slightly in the northern hemisphere. Although the general behavior of the level of no motion and the pycnocline scale coincides, the pycnocline scale $\eta_\rho$ measures how density stratification unfolds over the ocean column. In contrast, the level of no motion is a single layer at a certain depth. By their nature, the level of no motion $\eta_\psi$ and pycnocline
scale $\eta_\rho$ do not match exactly. Even though we know how density stratification unfolds, it is unclear whether the pycnocline scale represents the differences in stratification which are related to different transports. Considering the wind forcing dependencies, the pycnocline scale $\eta_\rho$ deepens in the 2XSH and 2X experiments relative to the 1X reference experiments throughout the basin. The pycnocline scales $\eta_\rho$ in the 2XSH and 2X experiments are congruent, and $\eta_\rho$ is mainly determined by the wind stress over the Southern Ocean. Local changes in the wind forcing at lower latitudes and in the northern hemisphere do not



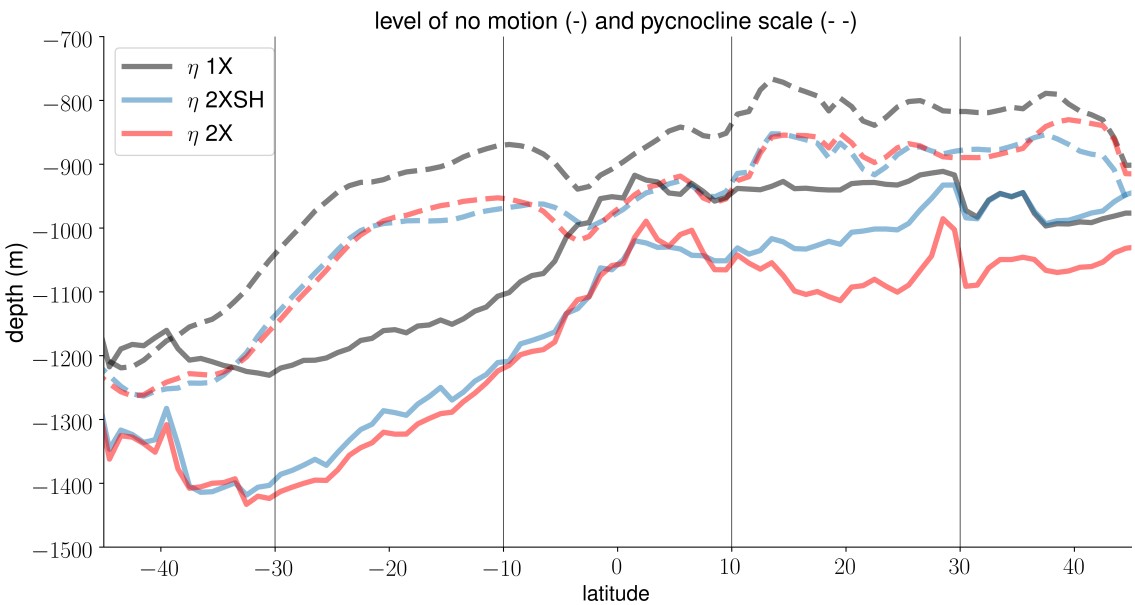

**Figure 4.** The time-mean (1991-2010) wind forcing dependencies of the level of no motion $\eta_\psi$ (solid) and the pycnocline scale $\eta_\rho$ (dashed) with respect to the 1X experiment (black), the 2XSH experiment (blue), and the 2X experiment (red).

change $\eta_\rho$ significantly. In the northern hemisphere, density stratification approximately unfolds at the same scale despite the differences in density stratification at the depth of $\eta_w$. In the southern hemisphere, the wind forcing dependence of the level of no motion $\eta_\psi$ is in line with the wind forcing dependence of pycnocline scale $\eta_\rho$. However, we find considerable differences between the wind experiments in the northern hemisphere due to an additional dependence on the local wind forcing. Nonlocal wind forcing over the Southern Ocean deepens the level of no motion $\eta_\psi$ in the northern hemisphere. Yet, in addition to this

nonlocal effect, a local effect acts on the level of no motion $\eta_\psi$ because we observe differences in the northern hemisphere subtropical region between the 2XSH and 2X experiments. The findings on the wind forcing dependencies of $\eta_\psi$ correspond to those findings on the differences in density stratification between the wind experiments.

### 3.3  Maximum overturning and its depth

Following the wind forcing dependencies of the level of no motion and pycnocline scale, we now analyze the wind forcing dependence of the northward flowing branch of the mid-depth cell. We compute the total maximum overturning streamfunction $\psi_t$ and the geostrophic maximum overturning streamfunction $\psi_g$. Conceptually, the differences between $\psi_t$ and $\psi_g$ provide insight on the degree to which the depth scale(s) are proxies for the strength of the AMOC. Computing the geostrophic maximum overturning streamfunction $\psi_g$, the level of no motion is unchanged, but the clockwise (upper) and counterclockwise (lower)

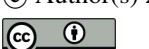



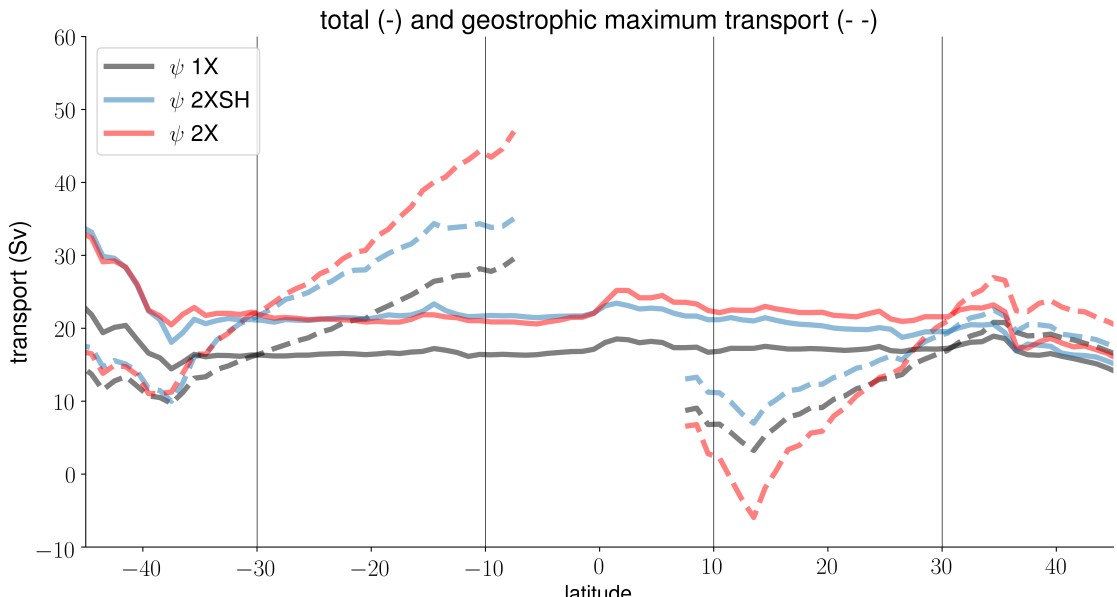

**Figure 5.** The time-mean (1991-2010) wind forcing dependencies of the total maximum overturning streamfunction $\psi_t$ (solid) and the geostrophic maximum overturning streamfunction $\psi_g$ (dashed) with respect to the 1X experiment (black), the 2XSH experiment (blue), and the 2X experiment (red).

rotating overturning cells are substantially altered. The latter puts into question the conservation of water masses, whereas the surface Ekman flux is independent of the conditions in the interior and does not change the relationship between the depth scales and northward transport at all. The maximum streamfunction $\psi_t$ includes the surface Ekman flux and the maximum streamfunction $\psi_g$ exludes the surface Ekman flux. However, the surface Ekman fluxes have to be compensated by an interior return flow that changes in relationship between overturning and its depth.


    Fig. 5 shows the meridional dependence of the total and geostrophic maximum overturning streamfunction ($\psi_t$,$\psi_g$). Considering $\psi_t$, the southern latitudes of the southern hemisphere are strongly influenced by the surface Ekman flux which scales with the zonal wind stress. Northward of the Southern Ocean (30S-10S), the AMOC becomes increasingly geostrophic. In the southern hemisphere, the strength of the mid-depth cell increases with higher wind forcing, and we find a stronger transport in
the 2XSH and 2X experiment than in the 1X experiment. In this region, the total maximum streamfunctions $\psi_t$ of the 2XSH and 2X experiments are approximately equal. However, $\psi_t$ intensifies north of the equator. In the subtropical region (10N-30N), $\psi_t$ increases with higher wind forcing, and the integrated transport of the 2X experiment is stronger than the integrated transport of the 2XSH experiment. In contrast to $\psi_t$, the geostrophic maximum overturning streamfunction $\psi_g$ is relatively weak near the Southern Ocean but constantly increases towards the equator. North of the Southern Ocean (30S-10S), $\psi_g$ is increasingly

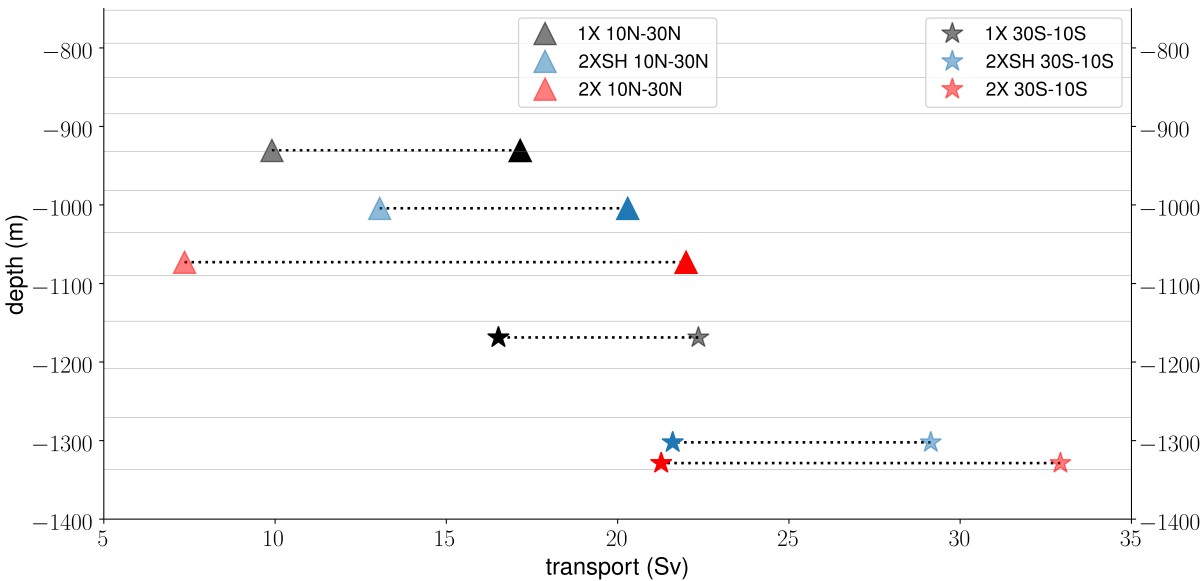

**Figure 6.** The time-mean (1991-2010) relationship between maximum overturning and its depth in the southern hemisphere and the northern hemisphere. The markers represent the meridional averages (30S-10S) (stars) and (10N-30N) (triangles). We show both the total maximum overturning streamfunction ($\psi_t$, opaque) and the geostrophic maximum overturning streamfunction ($\psi_g$, transparent). We do not show the pycnocline scale $\eta_\rho$ because the qualitative behavior is same as the behavior of the level of no motion $\eta_\psi$ except that the pycnocline scales of the 2XSH and 2X experiment are congruent throughout the basin. The thin grey lines indicate the vertical model grid.

dependent on the geostrophic return flow that compensates the southward surface Ekman flux locally. As a result of the interior return flow, $\psi_g$ becomes stronger with higher wind forcing, with markedly higher values in the 2X experiment compared the 2XSH experiment. We exclude the equatorial band where the approximation of the surface Ekman flux breaks down. At the low and mid-latitudes of the northern hemisphere (10N-30N), the interior return flow of the surface Ekman flux is directed southward and strongest at roughly 15°N. At these latitudes, $\psi_g$ of the 2X experiment is weaker than both the geostrophic

transport of the 1X reference experiment and the 2XSH experiment. We discuss the possible explanations for these differences in maximum overturning besides the Ekman return flow later on, in section 4. First, we highlight the relationships between the maximum overturning streamfunctions and the depth scales in order to bring together their spatial dependencies.

Combining our findings from Fig. 4 and 5, we describe the relationship between northward overturning and its depth from

a more nonlocal perspective on hemispheric differences. To highlight hemispheric differences in the inter-hemispheric region, we show the meridional averages (30S-10S) and (10N-30N) (Fig. 6). In the southern hemisphere, the level of no motion $\eta_\psi$ of the 2XSH and 2X experiment fall into the same model layer, and the associated maximum streamfunctions $\psi_t$ are not notably



different. In this region, $\psi_g$ increases with higher wind forcing. South of the equator, the wind forcing dependencies of the level of no motion $\eta_\psi$ as well as the pycnocline scale $\eta_\rho$ (not shown) are in line with the wind forcing dependencies of $\psi_t$, in

the sense that a broader depth scale is related to stronger northward overturning. Insights on the northern hemisphere are not as simple as in the southern hemisphere. The level of no motion $\eta_\psi$ deepens with higher wind forcing due to the downward displacement of isopycnals at mid-depth, whereas $\eta_\rho$ (not shown) reveals the same wind forcing dependencies as in its southern counterpart. The differences in the depth of the level of no motion $\eta_\psi$ between the wind experiments are important although they are set by one model layer only, because even a small change in the model layers implies a change in the accumulation

of vertical velocity shear, and the pycnocline scale $\eta_\rho$ does not account for these details. As in the southern counterpart, the differences in the level of no motion $\eta_\psi$ do not represent the changes in $\psi_g$, but the changes in $\psi_t$ are consistent with the common assumption that the northward transport of the mid-depth cell becomes stronger with a deeper level of no motion $\eta_\psi$. In general, the differences between $\psi_t$ and $\psi_g$ in the inter-hemispheric region suggest that the interior return flow of the surface Ekman flux is mostly compensated above the level of no motion $\eta_\psi$. In this regard, the level of no motion is a proxy for $\psi_t$

rather than a proxy for $\psi_g$. This raises the questions how well the total and geostrophic maximum overturning streamfunctions represent interior flow despite the surface Ekman flux or its geostrophic return, and how well the level of no motion and the pycnocline scale represent transport at different depths. In order to answer these questions, we analyze meridional velocity profiles ($\frac{\partial \psi}{\partial z}$) and shear ($\frac{\partial^2 \psi}{\partial z^2}$) from a more local and nonlocal perspective. That is, we analyze the meridional dependence of $\frac{\partial \psi}{\partial z}$ and meridional averages with respect to hemispheric differences.

## 4   The relationship between the depth scales and velocity profiles


The depth scales can be related to vertical velocity profiles ($\frac{\partial \psi}{\partial z}$) and shear ($\frac{\partial^2 \psi}{\partial z^2}$). To discuss changes in the velocity profiles with respect to the level of no motion and the pycnocline scale, we first compute the vertical derivative of the AMOC streamfunction $\frac{\partial \psi}{\partial z}$ (the zonal-mean meridional velocity scaled by the basin width). Neglecting the surface Ekman layer and integrating from the level of no motion $\eta_\psi$ to the top gives the geostrophic approximation of the maximum overturning streamfunction that

approximately represents the geostrophic conditions in the interior. Analyzing $\frac{\partial \psi}{\partial z}$ is a perspective on force balance, and we disentangle different contributions.

Fig. 7 shows the zonal-mean meridional velocities ($\frac{\partial \psi}{\partial z}$) in the wind experiments and the difference in $\frac{\partial \psi}{\partial z}$ between these wind experiments. The differences in $\frac{\partial \psi}{\partial z}$ between the wind experiments are strongest near the equator at the upper levels

where the vertical velocity shear changes drastically. Taking the difference between the 2X and 1X experiments, we find an increase in $\frac{\partial \psi}{\partial z}$ south of the equator and a decrease north of the equator. To a substantial extent, these changes can be attributed to the strengthening of the local Ekman cells. The differences in $\frac{\partial \psi}{\partial z}$ at the upper levels between the different experiments demonstrate that the Ekman return flow is baroclinic and occurs mostly above $\eta_\psi$. The strong influence of the Ekman cells near the surface suggests that, at these levels, the external wind-driven flow associated with the Ekman cells superposes the

internal flow that is associated with the level of no motion. These considerations support the perspective that $\eta_\psi$ is a proxy





**Figure 7.** The time-mean (1991-2010) vertical derivative of the AMOC streamfunction $\frac{\partial \psi}{\partial z}$ in the (a) 1X experiment, (b) 2XSH experiment, and (c) 2X experiment. We further show the difference in $\frac{\partial \psi}{\partial z}$ between the (d) 2XSH and 1X experiments, (e) the 2X and 1X experiments, and (f) the 2X and 2XSH experiments. The black lines represent the level of no motion $\eta_\psi$. We exclude the surface Ekman layer.



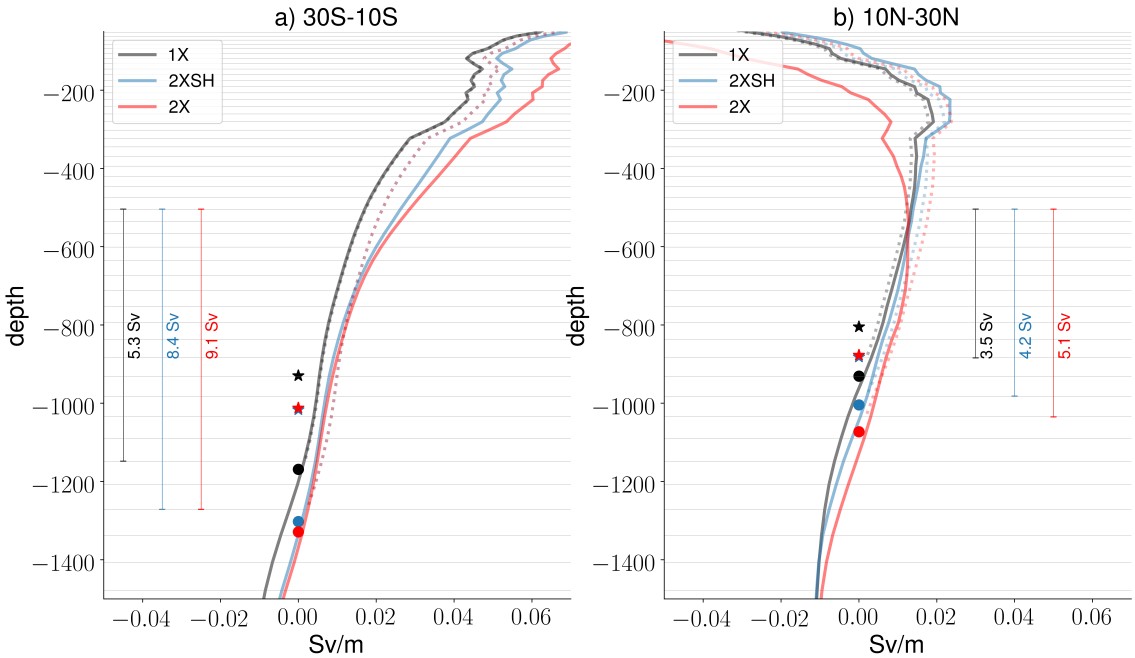

**Figure 8.** The meridional averages in the southern hemisphere (a, 30S-10S) and in the northern hemisphere (b, 10N-30N) of the time-mean (1991-2010) vertical derivative of the AMOC streamfunction $\frac{\partial \psi}{\partial z}$ in the 1X experiment (black), 2XSH experiment (blue), and 2X experiment (red). We exclude the surface Ekman layer. The circles show the meridional averages of the level of no motion $\eta_\psi$ and the stars show the meridional averages of the pycnocline scale $\eta_\rho$. The dotted lines represent the velocity profiles that arise from the displacements of the level of no motion $\eta_\psi$ while the velocity shear $\frac{\partial^2 \psi}{\partial z^2}$ is held constant (1X). We indicate the transport (Sv) at the deeper levels by the annotation at the vertical bars. The thin grey lines indicate the vertical model grid.

for $\psi_t$ rather than a proxy for $\psi_g$. Small differences in $\psi_t$ emerge in case of weak compensation of the surface Ekman flux below $\eta_\psi$. We further find differences in $\frac{\partial \psi}{\partial z}$ at mid-depth that are associated with the nonlocal wind forcing over the Southern Ocean. These changes can be inferred from the difference between the 2XSH and 1X experiments. The difference between these experiments is related to the enhanced inflow from the southern hemisphere into the northern hemisphere, which in turn

can be related to the strengthening of the western boundary current with an increase in wind forcing. In the 2X experiment, however, the perturbation associated with the Ekman cells is stronger than the perturbation associated with the strengthening of the western boundary current. The Ekman cells south and north of the equator further obscure the influence of the local Ekman pumping velocity $W_E$ on stratification and transport at the advective depth $\eta_w$, which can be hardly identified in Fig. 7. The changes in the wind stress curl with wind forcing are translated into the changes of the level of no motion $\eta_\psi$, and next

we analyze explicitly how different depths of the level of no motion $\eta_\psi$ result in different velocity profiles.




In this connection, we analyze whether there is a direct relationship between the vertical velocity profiles described by $\frac{\partial \psi}{\partial z}$ and the changes in the level of no motion $\eta_\psi$ in the sense that the vertical velocity shear $\frac{\partial^2 \psi}{\partial z^2}$ stays approximately constant (Fig. 8). The differences in the zonal-mean meridional velocities $\frac{\partial \psi}{\partial z}$ below the surface Ekman layer should establish a direct relationship between the depth scales and northward transport at least at the deeper levels. We focus on the meridional averages 30S-10S and 10N-30N to highlight the relationship between the velocity profiles and the depth scales $\eta_\psi$ and $\eta_\rho$. Deep transport has a substantial contribution in both hemispheres because shear accumulates in the vertical over a wide depth range, which is indicated by the vertical bars. In the southern hemisphere, the changes in the level of no motion $\eta_\psi$ and the pycnocline scale $\eta_\rho$ correspond to the changes in the actual velocity profiles $\frac{\partial \psi}{\partial z}$ (solid lines) at deeper levels. In the northern hemisphere downwelling region, the pycnocline scale does not change with local changes in wind forcing, whereas the level of no motion $\eta_\psi$ deepens locally. To assess to which degree the level of no motion $\eta_\psi$ represents changes in the velocity profiles, we vertically integrate the shear $\frac{\partial^2 \psi}{\partial z^2}$ of the 1X reference experiment from the level of no motion $\eta_\psi$ with zero reference velocity (dotted lines). This is not possible in the case of the pycnocline scale $\eta_\rho$, because it is a scale height and even a small layer difference in the reference depth results in differences in $\frac{\partial \psi}{\partial z}$. We find that the changes in deep velocity profiles are related to the changes in the level of no motion $\eta_\psi$ in both hemispheres. To a considerable extent, even the velocity profile of the 2XSH experiment at the upper levels in both hemispheres is connected to the displacement of the level of no motion $\eta_\psi$. In the 2X experiment, however, the signal that arises from the interior return flow of the surface Ekman flux overcomes the signal that arises from the displacement of $\eta_\psi$. In the southern hemisphere at the upper levels, the velocities associated with the changes of the level of no motion $\eta_\psi$ are lower than the actual velocities in this experiment. In the northern hemisphere at the upper levels, the velocities associated with the changes of the level of no motion $\eta_\psi$ are much higher than the actual velocities.

## 5 The response to global warming

Finally, we compare the insights on the wind experiments and the outcome of a warming experiment in which we quadruple the preindustrial $CO_2$ concentration. First, we characterize the evolution of the zonal-mean density stratification and the pycnocline scale $\eta_\rho$ (Fig. 9a) as well as the evolution of the zonal-mean meridional velocities $\frac{\partial \psi}{\partial z}$ and the level of no motion $\eta_\psi$ (Fig. 9b). Afterwards, we use meridional averages of the first and second vertical derivatives of the AMOC streamfunction $\frac{\partial \psi}{\partial z}$ and $\frac{\partial^2 \psi}{\partial z^2}$ in order to analyze explicitly the relationship between northward transport and its depth (Fig 10).

In a similar manner to the wind experiments, we normalize the zonal-mean potential density to highlight the changes in stratification, $\frac{\rho_0 - \rho}{\rho_0}$, with the reference density $\rho_0 = 1025$ kg m$^{-3}$. The evolution of the AMOC cells in the warming experiments suggests a continuous weakening and shoaling of the mid-depth cell (Fig. 2). We are interested in the changes of the depth scales $\eta_\psi$ and $\eta_\rho$ and their relationship to northward transport on a decadal timescale and do not consider the deviations from the model control state. To illustrate the temporal changes in density stratification, we first take the difference between 50-year multi-decadal means that split the 100-year model time series. Fig. 9a shows that density stratification changes throughout the mid-depth cell. We find the maximum change in density at an advective depth scale $\eta_w$ at the southern latitudes of the southern



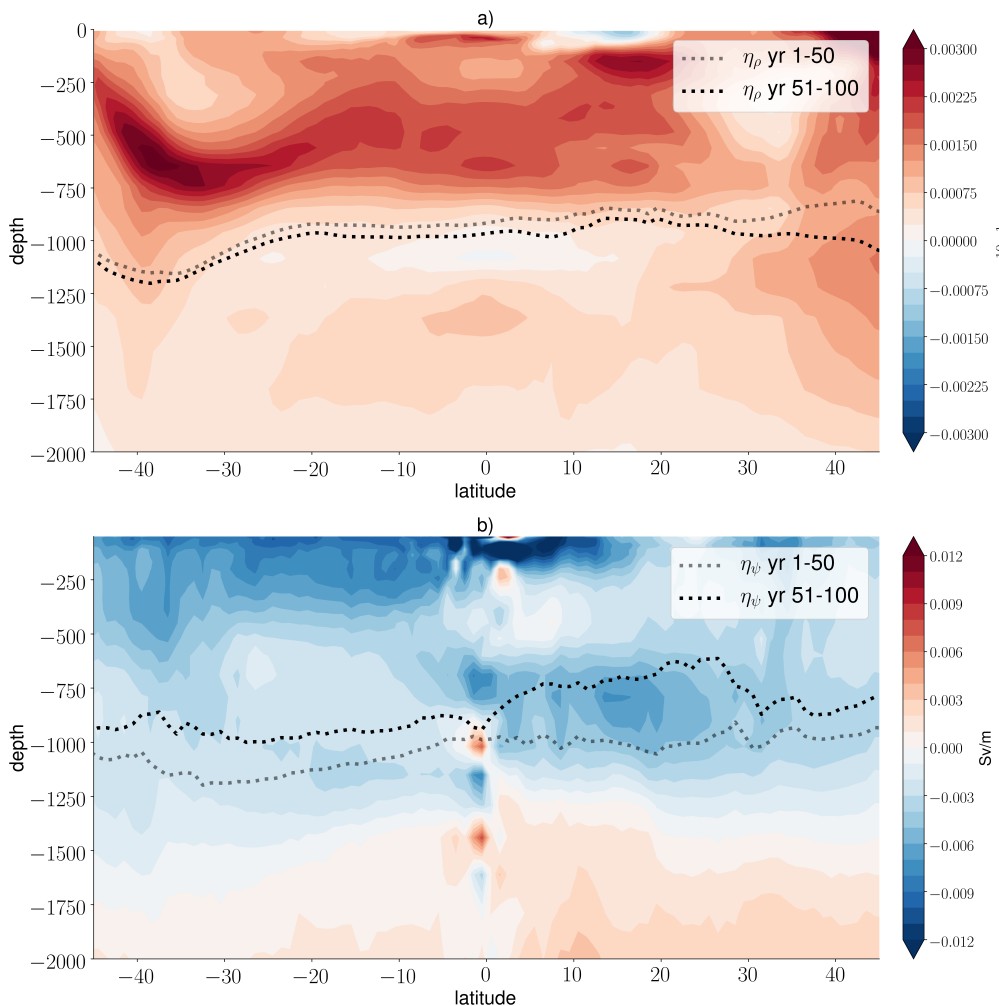

**Figure 9.** The difference in (a) the zonal-mean density stratification and (b) the vertical derivative of the AMOC streamfunction $\frac{\partial \psi}{\partial z}$ between years 51-100 and 1-50 in the 4XCO2 experiment. Density stratification is computed by $\frac{\rho_0 - \rho}{\rho_0}$, with the reference density $\rho_0 = 1025$ kg m$^{-3}$. The lines in (a) indicate the pycnocline scale $\eta_\rho$ and the lines in (b) indicate the level of no motion $\eta_\psi$.





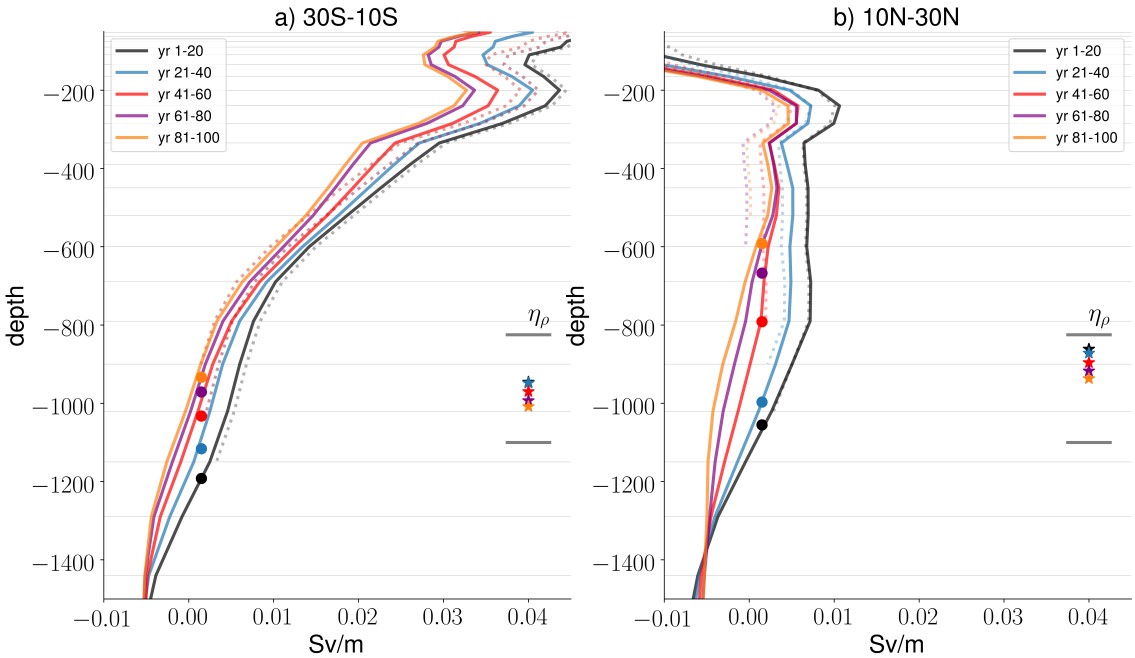

**Figure 10.** The meridional averages in the southern hemisphere (a, 30S-10S) and in the northern hemisphere (b, 10N-30N) of the vertical derivative of the AMOC streamfunction $\frac{\partial \psi}{\partial z}$ (solid) as a function of time in the 4XCO2 experiment. We take 20-year temporal means. We exclude the surface Ekman layer. The circles show the meridional averages of the level of no motion $\eta_\psi$ and the stars show the meridional averages of the pycnocline scale $\eta_\rho$ (aside). The dotted lines represent the velocity profiles that arise from the changes in the level of no motion $\eta_\psi$ while the velocity shear $\frac{\partial^2 \psi}{\partial z^2}$ is held constant (yr 1-20). The thin grey lines indicate the vertical model grid.

hemisphere (see Fig. 3), which indicates that wind forcing strongly influences ocean heat uptake in the Atlantic. The spatial distribution of heat uptake compensates for spatial variations of the pycnocline scale $\eta_\rho$. Heat diffuses downward even far below the level of no motion and is redistributed by the overturning circulation. As a result, the pycnocline scale $\eta_\rho$ broadens or deepens. In contrast to the pycnocline scale $\eta_\rho$, the level of no motion $\eta_\psi$ of the mid-depth cell shoals, and the meridional velocities $\frac{\partial \psi}{\partial z}$ of the upper branch of the mid-depth cell decrease (Fig. 9b). As in the case of the pycnocline scale, the spatial

variations of $\eta_\psi$ are less strong than the spatial variations in the different wind experiments, but they are still important considering the temporal adjustment. Furthermore, the decline in the magnitude of $\frac{\partial \psi}{\partial z}$ is evenly distributed in the latitude-depth section near or above $\eta_\psi$, which suggests that the displacement of the level of no motion changes the region over which vertical shear is accumulated. The perspective on constant deep shear ($\frac{\partial^2 \psi}{\partial z^2}$) is supported by the fact that we find a decrease in the zonal-mean meridional velocities even below $\eta_\psi$. Moving to deeper layers of the southward flowing branch indicates the

adjustment or weakening of the overturning circulation.





We highlight explicitly the linkage between velocity profiles ($\frac{\partial \psi}{\partial z}$), shear ($\frac{\partial^2 \psi}{\partial z^2}$) and the level of no motion $\eta_\psi$ with respect to the weakening of the mid-depth cell. Fig. 10 shows the meridional averages of the meridional velocity profiles $\frac{\partial \psi}{\partial z}$ and the depth scales (30S-10S, 10N-30N). We take 20-year temporal means to analyze the evolution of $\frac{\partial \psi}{\partial z}$ and the depth scales $\eta_\psi$ and $\eta_\rho$.

Similar to the level of no motion, the transport below the surface Ekman layer of the mid-depth cell decreases throughout the column and constantly with time (solid lines). The pycnocline scale $\eta_\rho$ behaves opposite to $\eta_\psi$ and does not scale the changes in the velocity profiles or northward transport. We do not find hemispheric differences in the qualitative behavior of the related quantities. To assess to which degree the changes in the level of no motion $\eta_\psi$ represent changes in $\frac{\partial \psi}{\partial z}$, we take the mean of the first twenty years of the warming experiment as reference profile and vertically integrate the shear $\frac{\partial^2 \psi}{\partial z^2}$ from the variation of

the level of no motion $\eta_\psi$ with zero reference velocity (dotted lines). We find that the evolution of the deep velocity profiles $\frac{\partial \psi}{\partial z}$ is directly related to the evolution of the level of no motion $\eta_\psi$. Below the surface Ekman layer the signal associated with the local wind forcing does not overcome the signal associated with altered surface buoyancy fluxes. However, the vertical velocity shear at the upper layers is a function of time, and the low vertical resolution of the model grid may not capture the details of the relationship between $\frac{\partial \psi}{\partial z}$ and $\eta_\psi$. The hemispheric differences in the shoaling of the level of no motion and transport

weakening suggest that the adjustment of the AMOC in response to global warming is not independent of location, and the level of no motion $\eta_\psi$ is an indicator for the transport weakening in the ocean column of the mid-depth cell.

## 6 Discussion

Our results justify the use of the wind experiments that are designed to explore the wind forcing dependence of the mid-depth cell south and north of the equator. In line with the current understanding of the Atlantic circulation, Southern Ocean winds

boost the strength of the AMOC and change density stratification throughout the basin (e.g. Vallis, 2000; Klinger et al., 2003, 2004; Klinger and Cruz, 2009). Northern hemisphere winds over the downwelling region additionally influence the meridional flow and density stratification locally, which is commonly ignored in the scientific literature on the AMOC. The present study is based on simulations with an eddy-resolving OGCM on a decadal timescale rather than a fully equilibrated experiment. We find a robust adjustment of the AMOC and density field, which demonstrates the realization of major adjustments due to wave

propagation. Analyzing the decadal response of the AMOC to Southern Ocean wind forcing, Klinger and Cruz (2009) show that the AMOC in the source region adjusts on an inter-annual timescale and the AMOC in the northern hemisphere adjusts on a decadal timescale. The adjustment timescale of relevant components such as the ACC or the pycnocline scale may depend sensitively on the experimental set-up and configuration of an OGCM, and in this connection the adjustment timescale may range from being multi-decadal to being centennial. For instance, Allison et al. (2011) outline theoretically that the inclusion of

North Atlantic deep water formation substantially shortens the timescale on which the Southern Ocean equilibrates in response to wind forcing. Considering the adjustment of the AMOC and density field in the wind experiments, the 30-year simulations are long enough to analyze the wind forcing dependencies of the depth scales and northward transport. Finally, we do not analyze the influence of wind forcing in the northern hemisphere upwelling region on the inter-hemispheric cell, because the experimental strategy of the present analysis is inappropriate to generate evidence on changes in the inter-hemispheric





circulation that arise from subpolar surface winds. Cessi (2018) finds a weakening of the inter-hemispheric cell in response
to increased westerly wind stress. Contrary to salt advection by the subpolar gyre, the return flow of the surface Ekman flux
opposes the sinking necessary to maintain the strength of the AMOC. According to our experimental strategy, the wind forcing
over the Southern Ocean and the wind forcing at the lower and mid-latitudes of the northern hemisphere predominantly set the
strength of the upper branch of the mid-depth cell in the region considered here.


     The findings of the present study support the pycnocline model described in Gnanadesikan (1999) in the sense that Southern
Ocean wind forcing deepens the pycnocline scale and the level of no motion and strengthens the AMOC. However, local
wind forcing over the northern hemisphere downwelling region additionally influences the level of no motion and northward
transport locally. In that respect, the level of no motion is more appropriate to scale northward transport than the pycnocline
scale. By artificial modification of density gradients in OGCM experiments, Griesel and Maqueda (2006) and DeBoer et al.
(2010) indicate that the pycnocline scale does not scale northward transport at all. By contrast, we provide insight on the
scaling behavior of the depth scales from a conceptual point of view, and the pycnocline scale does not account for local wind
forcing away from the Southern Ocean. Furthermore, the temporal adjustment of the AMOC in response to global warming
presumably differs between the southern hemisphere and the northern hemisphere. Levang and Schmitt (2019) address the
northern hemisphere weakening of the AMOC in global warming simulations and analyze temperature-induced and salinity-
induced changes in the shear component of the meridional flow. In this connection, their findings are in line with the finding
that local Ekman pumping influences the mid-depth cell in a substantial way. Finally, Levermann and Fuerst (2010) evaluate
the pycnocline model and show that meridional density gradients and the pycnocline scale are mutually independent. They use
a model of intermediate complexity and analyze equilibrated experiments which reproduce the response to Southern Ocean
wind forcing and global warming, among others. According to their findings, the pycnocline scale and meridional density
gradients vary in the case of the wind experiments. Considering their global warming experiments, the pycnocline scale varies
while meridional density gradients are fixed. Using meridional density gradients instead of zonal density gradients is based on
the assumption that these gradients are proportional and have the same order of magnitude, and zonal and meridional velocities
compare well with one another. Using the level of no motion as depth scale on a decadal timescale, the present study shows
that changes in deep meridional velocity profiles are directly related to changes in the level of no motion because deep velocity
shear and thus zonal density gradients remain approximately constant. Comparing the wind experiments, the ocean response at
the upper levels is much more complex than the response at the deeper levels, which is mostly related to the baroclincity of the
interior return flow of the surface Ekman flux. Supporting the considerations made in McCreary and Lu (1994) and Klinger and
Marotzke (2000), our findings strongly suggest baroclinic Ekman compensation which has been demonstrated in an idealized
way by Williams and Roussenov (2014). The displacement of the level of no motion in the MPIOM experiments approximates
the conditions in the interior with the Ekman cells mainly cancelled out.



## 7 Summary and Conclusions

We use wind sensitivity experiments to analyze the depth scales of the inter-hemispheric cell in the Atlantic and their relationship to northward transport. We focus on the inter-hemispheric region in order to analyze the interplay of nonlocal and local wind effects, and our perspective deviates from the common view that the AMOC is a nonlocal phenomenon only. The dynamics of the inter-hemispheric cell can only be understood by analyzing both Southern Ocean wind effects and local wind effects in the northern hemisphere downwelling region which arises from the forcing imposed by the wind stress curl.

We find different wind forcing dependencies of the pycnocline scale and the level of no motion. Southern Ocean processes determine the magnitude of the pycnocline scale, whereas northern hemisphere wind stress additionally influences the level of no motion. The pycnocline scale is insensitive to the local wind stress over the northern hemisphere and cannot capture the details of deep velocity profiles and mid-depth stratification. Local wind forcing changes density stratification and displaces isopycnals downward at an advective depth. In that respect, the level of no motion is a better proxy for velocity profiles than the pycnocline scale, because the level of no motion accounts for the changes in the surface winds over the northern hemisphere.

To a large extent, the changes in deep transport below the surface layers between the wind experiments can be related to the changes of the level of no motion in the case that we hold the vertical velocity shear constant. However, in the wind experiments the representation of velocity profiles by the level of no motion is restricted to the deeper layers and not directly coupled to the strong geostrophic flow near the surface, which is attributed to the presence of the Ekman cells. In this regard, the changes in the meridional flow which are associated with the displacement of the level of no motion and the actual transport can differ considerably at these levels. Our findings indicate that the baroclinic return flow of the surface Ekman flux occurs mostly above the level of no motion. Future research on the Ekman cells and the related changes in the density field will influence the way how we decompose the AMOC into different components.

There is no unique way to describe and quantify interior geostrophic flow that is not directly influenced by the local Ekman cells, and in this sense both the total maximum overturning streamfunction and the geostrophic maximum overturning streamfunction are approximations for the conditions in the interior. Our findings suggest that the differences in the total maximum overturning streamfunction are related to the differences in the level of no motion because the surface Ekman fluxes are compensated mostly above the level of no motion. Compared to the total maximum overturning streamfunction, the geostrophic approximation makes a scaling more complex, but it is the result of the force balance below the surface Ekman layer. The hemispheric differences in the level of no motion and the associated meridional transport suggest a hemispheric scaling rather than a single depth scale approximation for the entire basin. In the northern hemisphere the relationship between overturning and its depth is determined by the combination of nonlocal and local wind effects.



The global warming experiment shows that changes in the the level of no motion over time represent changes in northward transport in the upper branch of the mid-depth cell. That is, the deep vertical velocity shear stays approximately constant while the level of no motion shoals. Although the sign of the changes is equal in both hemispheres, we find hemispheric differences in the shoaling of the level of no motion and transport weakening. In terms of a direct relationship, the pycnocline scale cannot scale northward transport on the decadal timescale considered here, because it deepens after an initial adjustment. By contrast,

the pycnocline scale may be a time-dependent predictor for future long-term overturning in response to global warming because it adjusts due to advective-diffusive balance.

The present manuscript relies on experiments that are conducted with a single but horizontally high-resolution model. Low-resolution models may differ significantly from high-resolution models. We put forward the idea that the ability of numerical

models to capture the spatial and temporal variations of the level of no motion is crucial to reproduce the mid-depth cell in an appropriate way.

*Data availability.* Data will be published via MPG.PuRe. We publish the data associated with the present study during the external review process. The codes associated with the present study are available upon reasonable request.

*Author contributions.* TR conducted the research and developed the coding scripts associated with the present study. TR developed the text

of the present manuscript. JB, VL, DP, and JM reviewed the manuscript and checked the consistency of the research results.

*Competing interests.* The authors declare that they have no conflict of interest.

*Acknowledgements.* We thank Oliver Gutjahr for the internal review on the present manuscript. This work was funded by the Max Planck Society (MPG) and the International Max Planck Research School on Earth System Modelling (IMPRS ESM). JB received funding under Germany's Excellence Strategy, EXC 2037 'Climate, Climatic Change and Society' CLICCS, Project Number: 390683824, as contribution

to the Center for Earth System Research and Sustainability (CEN) of Universität Hamburg. DP received funding under PRIMAVERA, a Horizon 2020 poject funded by the European Comission, with grant number 641727. We also thank the German consortium project STORM for supporting the realization of high-resolution model simulations. We further thank Deutsches Klima Rechenzentrum (DKRZ) for providing the computational resources.



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
