# Peer review of "Nonlocal and local wind forcing dependence of the Atlantic meridional overturning circulation and its depth scale"

_Ocean Science, 2020_

## Referee Comment (RC1)

**Overview**

The paper examines a handful of numerical experiments to analyze the relationship between AMOC strength, the vertical structure of density and streamfunction, and external forcing such as wind. Most of the experiments are done at high resolution, and because of computational expense this necessitates short runs and hence conclusions only apply to decadal timescales. The paper attempts to isolate separate effects of Southern Ocean wind (associated with global-scale features) and northern hemisphere wind (associated with North Atlantic features only). It also attempts to link changes in the vertical scales to changes in volume transport.

The results are potentially interesting, but the writing is so difficult to penetrate that the manuscript needs a major revision before I can fairly judge the scientific content. I do worry that the small number of experiments and relatively light analysis are not able to support the conclusions of the paper about the relationship between depth scales and volume transports. However, maybe these issues will become clearer after a rewrite. I give more information about my problems with the writing in the sections below.

**1. Motivation of Work**

The paper is very poorly motivated. In the Introduction, the 1$^{st}$ paragraph is okay, the 2$^{nd}$ paragraph is a little dense, and the 3$^{rd}$ and final paragraph is very hard to read and does not do a good job of explaining why these particular experiments were done.

The 3$^{rd}$ paragraph of the Intro skips back and forth between mentioning subtropical gyre theory (Luyten et al), thermocline behavior due to diffusion and Southern Ocean wind stress (Vallis 2000), and northern high latitudes (Cessi, 2018). Three depth scales are mentioned, but the "advective depth" isn't defined. Wind stress curl (Cabanes et al, 2008) is mentioned, but again without much explanation. Reference is made to "local" versus "nonlocal" wind influence but its unclear if "local" refers to wind and thermocline thickness in the same hemisphere, wind and thermocline both in low/mid latitudes, or wind and thermocline in the same latitude band. For this reason the paper does not clearly state what question is being asked. It doesn't give motivation for the particular wind patterns used.

Intro also does not explain why an eddy-resolving model is used; this is important because the computational expense of an eddy-resolving model forces the experiments to be very short compared to the adjustment timescale of the large-scale thermocline to wind.

Finishing the Intro with only a vague idea of what the authors were trying to learn, I found the discussion of the results hard to follow as well.

**2. Sec 3.1 results confusing**

Figure 3 shows wind effects on isopycnals, as stated in para 2 of the section. However, paragraph states "displacement is scaled by $\eta_w$". How do we know this from the figure or from the experiments? Why do "We expect considerable changes in the level of no motion"?

After stating that its easy to see why changing the stratification changes the Level of No Motion, the paragraph gives an argument I can't follow. "displacement of the isopycnals... is small" because "change in deep vertical shear is presumably small". I don't see how one is related to the other, or why the change in deep shear is presumed small. And why do density differences at the advective depth imply changes in velocity at mid-depth? What is mid-depth – 2 km? The paragraph mentions that zonal pressure gradients are related to the Level of No Motion, but I don't see how that fact helps us since the paper doesn't present any information about the zonal pressure gradients.

**3. Some Sec 4 results not new?**

Sec 4 shows that the subtropical cells are largely confined to the top few hundred meters, which has long been known (see for instance McCreary & Lu, 1994, JPO). Para 2 asserts that this suggests that "$\eta\_\psi$ is a proxy for $\psi_t$ rather than... $\psi_g$". Doesn't it show the opposite? If the Ekman cells are above $\eta_\psi$, shouldn't their transport be independent of $\eta_\psi$?

The last paragraph of the section seems to say that if we assume that the vertical shear is constant (why should it be?), than changes in $\eta_\psi$ produce changes in volume transport. However, it is unclear to me if the paper actually shows that. Again, the difficult writing style makes it harder to tell.

**4. Generally Opaque Writing Style**

In general, almost every sentence is difficult to read. The authors add words to the sentences that do not give any information, and they omit words that would specify what they are talking about. Often the connection between one sentence and the next is not clear. Below, I give some examples of difficult paragraphs. Unfortunately, many paragraphs have similar problems.

It would help if paragraphs were not so long. For instance in the 2nd paragraph of the Intro, "The depth scale itself is determined..." can start a new paragraph, "Different assumptions like..." can start a new paragraph, and "The present study addresses" can start a new paragraph. Similarly, paragraph 3 can be divided into 3 or 4 paragraphs, as can many other paragraphs throughout the paper. Ideally each paragraph gives one main idea, otherwise its easy for the reader to get lost in overlong blocks of text.

**Examples of Difficult Paragraphs**

Original text in black. Red text refers to **boldface** parts or, if no boldface, the entire sentence.

**(a) From Abstract:**
Our findings deviate from the common perspective that the AMOC is a nonlocal phenomenon only, because northward transport in the inter-hemispheric cell **can only be understood by analyzing** nonlocal Southern Ocean wind effects and local wind effects **in the northern hemisphere downwelling region** where Ekman pumping takes place.

This is very vague – what about the transport can only be understood? What information does analyzing the effects give us?

The experiments compare runs with winds differing over the entire basin N of 30S, so how do we isolate effects of wind in downwelling region alone?

Southern Ocean wind forcing predominantly determines the magnitude of the pycnocline scale throughout the basin, whereas northern hemisphere winds additionally influence the **level of no motion**  locally.

OK, except **level of no motion** (LoNM) is used in a nonstandard way here. In physical oceanography, LoNM refers to a depth where velocities are close to zero.  In this paper, it refers to a depth where the zonal average velocity is zero.  There may be strong velocities that cancel out in the zonal average.  Its okay to use LoNM this way but could be confusing in Abstract.

In that respect, the level of no motion is a better proxy for northward transport and mid-depth velocity profiles despite the Ekman return flow which is found to be baroclinic.

This sentence is trying to discuss so many issues at once that the reader can not understand any of them.

**We compare our results inferred from the wind experiments and a** 100-year global warming experiment in which the atmospheric CO2 concentration is quadrupled, using MPIOM coupled to an atmospheric model.

Mentioning the comparison to the wind experiments just makes the sentence longer without adding information.

We find that the evolution of the level of no motion in response to global warming **represents changes in vertical velocity profiles or northward transport**, whereas the changes of the pycnocline scale are opposite to the changes of the level of no motion over time.

Don't know what that means. Represents in what way?  What changes in profiles or transport? Maybe should just combine last 2 sentences, like "A 100-year global warming experiment shows that the pycnocline depth scale increases and the depth of the LoNM decreases."  Note that I've written a simple, understandable sentence.  The paper needs more of those!

Using the level of no motion as depth scale, the analysis of the wind experiments and the warming experiment suggests a hemisphere-dependent scaling of the strength of AMOC.

I don't understand how the words before the comma (",") connect to the words after the comma. AMOC strength depends on depth of LoNM? In a particular hemisphere?

Furthermore, we put forward the idea that the ability of numerical models to capture the spatial and temporal variations of the level of no motion is crucial to reproduce the **mid-depth cell** in an **appropriate way**.

What is that?  The mid-depth flow is part of a top (surface to NADW) cell and a bottom (NADW to AABW) cell.

 What is meant by that?  Accurate? Dynamically correct?

**(b) From Sec 2.2, para 1**

We analyze the AMOC in the **inter-hemispheric region** south and north of the equator.

What region is that? Between 30S and 30N? Between 30S and 60N?

In this way, we explore the nonlocal response to changes in Southern Ocean winds and local wind effects in the **downwelling region** of the northern hemisphere.

How does studying the unspecified inter-hemispheric region tell us specifically about the downwelling region?

In general, the mid-depth cell strengthens with higher wind forcing over the Southern Ocean, but **we cannot** capture **the details** of the different experiments **in terms of spatial variations**.

Should tell readers what the study can do, not what it can't do. This is especially confusing because it seems that the point of the study is to capture the spatial variations.

Vague. The experiments have lots of details, but the paper is only about some of them.

Is this referring to spatial variations in the wind forcing, the transport, or both?

The surface meridional Ekman flux can be inferred from the surface levels of the overturning streamfunction; it scales with the zonal wind stress and is inversely proportional to the Coriolis parameter.

True, but why is this mentioned here?

In contrast to the surface Ekman flux which is negative south of the equator and positive north of the equator, the northward flow of the mid-depth cell seems to be continuous and contiguous throughout the basin, **and it is difficult to base inferences on purely regional dynamics.**

Inferences about what? Do you want to know if the subtropical Ekman transport influences the strength of the AMOC?

However, these surface Ekman fluxes already indicate that the flow is not as continuous as the AMOC streamfunction suggests, because they have to be compensated by an interior return flux which changes the force balance of the flow.

If the point of the experiments is to understand how the subtropical Ekman transport affects the AMOC, that should be stated in the Introduction, not in the middle of this paragraph.

Furthermore, the wind stress curl over the basin imposes a forcing that may change stratification locally, **in the sense** that the meridional transport and its depth **differ between the wind experiments.**

This phrase generally implies an equivalence, but change in stratification is a separate issue from change in meridional transport. This phrase is used in several places in the paper and probably should be replaced in all cases by something clearer.

Between which wind experiments? Paragraph was talking about subtropical wind, but no experiment discussed here only changes subtropical wind.

In the 4XCO2 experiment, the surface buoyancy fluxes change continuously.

I thought the paragraph was about wind variations. I still don't understand the connection between the Ekman issues described here and the papers' experiments, but now we have switched to talking about buoyancy fluxes. Why?

On a multi-decadal timescale, the mid-depth cell weakens and shoals after the forcing is switched on (Fig. 2d,e,f).

So what?

The 100-year simulation time series makes it possible to analyze multi-decadal changes and compare the 4XCO2 experiment and the wind experiments.

Is this sentence trying to justify doing a 4XCO2 experiment, or doing the simulation for 100 years? Why do we want to compare a 4XCO2 experiment with wind experiments?

The warming experiment provides conceptual understanding of the linkage between ocean heat uptake and changes in the depth scales of the AMOC and how they relate to northward transport.

This thought should come at the beginning of a paragraph, preferably in the Intro. You are hypothesizing a connection between depth scale and transport, and 4XCO2 is useful because both change?

We summarize the experimental strategy as outlined above in Table 1.

**5. Small Issues**

**Sec 1 para 3:** Paragraph emphasizes lack of work on effects of northern hemisphere wind on overturning, but Klinger et al (2003) and (2004) both looked a effect of Westerlies in both hemispheres.

**Sec 2.2 para 1:** Definition of streamfunction should be indefinite integral in z, not definite integral. Otherwise $\psi$ won't be a function of $z$.

---

## Author Comment (AC3)

**Rohrschneider et al. 2021, response letter**

**The depth scales of the AMOC on a decadal timescale**

With the authors' response we re-initiate the publication process and soon submission of the revised manuscript. We are grateful for the many comments which have been posted by the anonymous referees. In the following we outline the future changes on the present manuscript. Our study is based on two sets of experiments only but we believe it will be a major contribution to the scientific community because the local wind forcing dependence of the AMOC has not been explored yet.

We changed the manuscript substantially. We focus on the wind experiments only and neglect the global warming experiment. The content of the paper is now well summarized by the abstract.

We use wind sensitivity experiments to understand the wind forcing dependencies of the level of no motion as the depth of maximum overturning and the e-folding pycnocline scale as well as their relationship to northward transport of the mid-depth Atlantic meridional overturning circulation (AMOC). In contrast to previous studies, we investigate the interplay of nonlocal and local wind effects on a decadal timescale. We use 30-year simulations with a high-resolution ocean general circulation model (OGCM) which is an eddy-resolving version of the Max Planck Institute Ocean Model (MPIOM). Our findings deviate from the common perspective that the AMOC is a nonlocal phenomenon only, because northward transport and its depth scales depend on both nonlocal Southern Ocean wind effects and local wind effects in the northern hemisphere downwelling region where Ekman pumping takes place. Southern Ocean wind forcing predominantly determines the magnitude of the pycnocline scale throughout the basin, whereas northern hemisphere winds additionally influence the level of no motion locally. In that respect, the level of no motion is a better proxy for northward transport and mid-depth velocity profiles than the pycnocline scale, since the wind forcing dependencies of the level of no motion and maximum overturning are equal. The changes in maximum overturning with wind forcing are explained by the changes in the level of no motion only. This is because wind-driven Ekman compensation is baroclinic and occurs above the level of no motion, and the internal vertical velocity shear that is not influenced by the external Ekman cells stays approximately constant. The analysis of the wind experiments suggests a hemisphere-dependent scaling of the strength of AMOC. We put forward the idea that the ability of numerical models to capture the spatial and temporal variations of the level of no motion is crucial to reproduce the mid-depth cell in an appropriate way both quantitatively and dynamically. (line 1-17)

We answer explicitly the following research questions.

This paper presents an analysis of wind sensitivity experiments in order to provide insight into the wind forcing dependence of the inter-hemispheric circulation by understanding the behavior of the depth scale(s) of the AMOC. (line 28-30)
…
Understanding the wind forcing dependence of the AMOC by understanding its depth scales makes the underlying research question twofold, in the sense that we discuss the wind forcing dependence of the AMOC using the depth scales and we discuss whether the depth scales are proxies for northward transport to understand the wind forcing dependence. We hypothesize that the level of no motion is a proxy for northward transport in the inter-hemispheric cell because the background vertical velocity shear of the meridional velocity may stay constant under changing wind forcing. (line 62-67)

…
We focus on the inter-hemispheric region 30S-30N and analyze the interplay of nonlocal wind forcing over the Southern Ocean and local wind forcing in the northern hemisphere downwelling region where Ekman pumping takes place. (line 70-71)
….
We address the question how changes in both nonlocal and local wind forcing influence the AMOC. We hypothesize that the influence of northern hemisphere winds on the AMOC is substantial. (line 85-86)

We changed the discussion accordingly and discuss the wind forcing dependence of the AMOC, the mechanism of the wind forcing dependence, and the depth scales as proxies for meridional

flow. Theses topics are not limited to the discussion but the major focus of the present manuscript. The paper is much more tailored regarding the focus of the present manuscript. In this connection, we did rewrite the introduction, say why our high-resolution simulation is necessary, elaborate on the mechanism of the wind forcing dependence with a new figure, inter alia.

We improved the writing considerably throughout the manuscript. In this response letter, we cannot state all changes made, but we made changes in every section in order to improve the flow of the paper.

Please review both response letters as they are entangled due to the sheer amount of the changes made.

**I)-V) : major comments**

**VI) : minor comments**

:*comment*

:response

**I) REFEREE 1**

*The paper is very poorly motivated. In the Introduction, the 1st paragraph is okay, the 2nd paragraph is a little dense, and the 3rd and final paragraph is very hard to read and does not do a good job of explaining why these particular experiments were done. The 3rd paragraph of the Intro skips back and forth between mentioning subtropical gyre theory (Luyten et al), thermocline behavior due to diffusion and Southern Ocean wind stress (Vallis 2000), and northern high latitudes (Cessi, 2018). Three depth scales are mentioned, but the "advective depth" isn't defined. Wind stress curl (Cabanes et al, 2008) is mentioned, but again without much explanation. Reference is made to "local" versus "nonlocal" wind influence but its unclear if "local" refers to wind and thermocline thickness in the same hemisphere, wind and thermocline both in low/mid latitudes, or wind and thermocline in the same latitude band. For this reason the paper does not clearly state what question is being asked. It doesn't give motivation for the particular wind patterns used.*

We spent much effort into the motivation and writing style of to improve the new paper. In fact, these two issues are entangled and the writing style strongly influences the perception of the reader about the motivation and definitions and arguments. The author's writing style is to write

in a condensed way. With the third paragraph we now try to synthesize the current research stage of the nonlocal and local wind forcing dependence of the AMOC, and we try to guide the reader through the research questions. Throughout the present study nonlocal wind forcing dependence means the dependence of the basin-wide AMOC on winds over the Southern Ocean and local wind forcing dependence means the dependence of the AMOC in the northern hemisphere on the surface winds over the northern hemisphere downwelling region where Ekman pumping takes place. Considering the wind forcing, we change the meridional and zonal velocities because we are especially interested into the local wind forcing dependence of the AMOC in the northern hemisphere downwelling region. The latter is related to the wind stress curl. We are adding explicit explanations. Based on our changes we state the research questions on the wind forcing dependence more clearly, and we state why we analyze the depth scales of the AMOC.

We propose to change the 2nd, 3rd, and 4th paragraph of the introduction in the following way:

review line 33-86

*Intro also does not explain why an eddy-resolving model is used.*

The OGCM used in the present study is a high-resolution eddy resolving MPIOM version (TP6ML80). From an overarching perspective, we simply expect a more realistic simulation compared to low-resolution MPIOM versions. In section *Experiments and Methods* (2) we mention that we assume better model physics in terms of the resolution of mesoscale eddies as well as wave propagation, compared to low resolution models. In this connection, a published study is lacking, but Gutjahr et al. (2019) already indicate that the high-resolution MPIOM configuration reveals the most realistic simulation compared to low-resolution MPIOM versions in terms of the mean state of the ocean. For instance, a cold bias in sea surface temperature over the Southern Ocean is strongly reduced, because the resolved eddies better influence the flattening and cropping of isopycnals compared to the GM thickness diffusivity parameterization. The latter is switched off. The authors realized a preliminary analysis of the effect of model resolution on the wind forcing dependence of the AMOC. They basically compared TP6ML80 with GR15L80 (1.5 degrees horizonal resolution and 80 vertical layers) and find that qualitatively the wind forcing dependence of the level of no motion is the same (the dependence on nonlocal Southern Ocean wind forcing and local wind forcing over the northern hemisphere downwelling

region). However, the horizontal transmission of density signals is sensitive to the horizonal model grid and the accumulation of vertical shear is sensitive to the vertical model grid, which inter alia changes quantitatively the relationship between the depth scales and meridional velocity profiles. Concerning the current state of model development, these details matter and are a source of uncertainty. We now outline in more detail the need of the high-resolution MPIOM version for our scientific study. We explain the advantage of the high resolution MPIOM version already in *Introduction* (1) now and then extend it in *Experiments and Methods* (2). Beyond the scope of the paper, we do not integrate a model inter-comparison.

We propose to change the 5[th] paragraph *Introduction* in the following way:

review line 88-96

We propose to change the 1[st] paragraph of *Experiments and Methods* in the following way.

review line 109-120

*The computational expense of an eddy-resolving model forces the experiments to be very short compared to the adjustment timescale of the large-scale thermocline to wind.*

As outlined in *Experiments and Methods* (2), the experimental burden of an eddy-resolving simulation is large and we therefore use 30-year wind experiments only. The forcing is switched-on at year 1980, and we analyze the time window 1991-2010 after the realization of major adjustments on a decadal timescale. The wind experiments are justified by the robust adjustment of the AMOC and density field on the timescale considered here. The experiments are not fully equilibrated but major adjustments are realized on a decadal timescale. That is, the wind experiments reflect the wind forcing dependence of the AMOC. For instance, the basin-wide transmission of density signals by wave propagation (e.g. Rossby waves) occurs on a pentadal and decadal timescale. The actual adjustment timescale of the pycnocline is longer and may depend sensitively on the experimental set-up. We now discuss the adequacy of the 30-year simulations in *Experiments and Methods* (2).

We propose to change the 2[nd] paragraph of *Experiments and Methods* in the following way.

review line 121-135

**II) REFEREE 1**

*Figure 3 shows the wind effects on isopycnlas, as stated in para 2 of the section. However, paragraph states "displacement is scaled by ηw". How do we know this from the figure or from the experiments? Why do "We expect considerable changes in the level of no motion"? After stating that its easy to see why changing the stratification changes the Level of No Motion, the paragraph gives an argument I can't follow. "displacement of the isopycnals... is small" because "change in deep vertical shear is presumably small". I don't see how one is related to the other, or why the change in deep shear is presumed small. And why do density differences at the advective depth imply changes in velocity at mid-depth? What is mid-depth – 2 km? The paragraph mentions that zonal pressure gradients are related to the Level of No Motion, but I don't see how that fact helps us since the paper doesn't present any information about the zonal pressure gradients.*

The maximum change in density between the wind experiments coincides with the advective depth scale. The latter is proportional to Ekman pumping and thereby to the wind stress curl which displaces isopycnals downward. We now state this explicitly as it is a matter of writing. Furthermore, we now provide some theoretical background such as thermal wind balance. In general, the mechanism how the level of the motion changes with wind forcing and how we understand the wind forcing dependence prevails throughout the new manuscript now. It is a major research question which is reflected in the abstract and introduction. We did rewrite the section on density stratification in order to meet the comments made by the referee. Our findings suggest that changes in zonal pressure gradients at the advective depth are related to significant changes in the level of no motion of the AMOC, because the vertical velocity shear stays approximately constant with wind forcing (formally a matter of mathematical integration). Section (4) closes the gap and explains explicitly how much of the changes in volume transport can be explained by the level of no motion in the case that we hold the vertical velocity shear constant. To illustrate, synthesize and state this more explicitly, we added a new paragraph at the end of section (4) and neglect theoretical arguments in the section on density stratification, section (3). We try to make better connections.

Besides changing the abstract and introduction as stated above, we propose to change section (4) in the following way. We neglect the mechanism is section (3).

Section (4):
In this connection, we analyze whether there is a direct relationship between the vertical velocity profiles described by $\frac{\partial {\psi}}{\partial {z}}$ and the changes in the level of no motion $\eta_\mathrm{\psi}$, by computing the meridional averages 30S-10S and 10N-30N (Fig. 8a,b). We analyze how much of the velocity profiles can be predicted by the level of no motion only. For this purpose, we assume vertical velocity shear that is constant with changing wind forcing. (line 341-344)

...

Integrating the velocity profiles vertically, however, the Ekman cells should cancel out such that the level of no motion is a proxy for northward transport.

We therefore show the changes in maximum overturning that are associated with the level of no motion only by analyzing the vertically integrated transport. Fig. 9 shows the total maximum overturning streamfunction $\psi_\mathrm{t}$ and the maximum overturning streamfunction $\psi^\mathrm{*}$ in the case that we hold the vertical velocity shear constant. We find that the changes in total maximum overturning $\psi_\mathrm{t}$ are explained by the changes in the level of no motion to a very large degree. The maximum overturning streamfunctions $\psi_\mathrm{t}$ and $\psi^\mathrm{*}$ are approximately congruent, and the vertical velocity shear $\frac{\partial^2 {\psi}}{\partial {z}^2}$ stays approximately constant with wind forcing on the timescale considered here. There is deviation at the equator due to systematic errors that arise from perturbations in equatorial upwelling. Away from the equator, however, the differences between the 2XSH and 2X experiments arise solely from the differences in the level of no motion. The mechanism how the changes in wind forcing translate into changes in the level of no motion $\eta_\mathrm{\psi}$ is thus easy to comprehend from a generic point of view. Small changes in the zonal pressure gradients at the depth range scaled by advective depth scale $\eta_\mathrm{w}$ are related to substantial differences in the level of no motion $\eta_\mathrm{\psi}$, because the internal velocity shear that is not influenced by the external Ekman cells hardly changes between the wind experiments. (line 358.-373)

**III) REFEREE 1**

*Sec 4 shows that the subtropical cells are largely confined to the top few hundred meters, which has long been known (see for instance McCreary & Lu, 1994, JPO). Para 2 asserts that this suggests that "ηψ is a proxy for ψt rather than... ψg". Doesn't it show the opposite? If the Ekman cells are above ηψ, shouldn't their transport be independent of ηψ? The last paragraph of the section seems to say that if we assume that the vertical shear is constant (why should it be?), than changes in ηψ produce changes in volume transport. However, it is unclear to me if the paper actually shows that. Again, the difficult writing style makes it harder to tell.*

Our study supports the perspective that the interior return flow of the surface Ekman flux is baroclinic. McCreary and Lu (1994) already show theoretically that the subtropical cells are baroclinic; i.e. the interior return flow of the surface Ekman flux is baroclinic. However, in the current literature on the AMOC it is not well established to assume a baroclinic interior return flow of the surface Ekman flux on the timescale considered here. For instance, as outlined in the paper, a range of studies assumes that the interior return flow of the surface Ekman flux is barotropic even on longer than monthly timescales. Using an idealized OGCM experiment, Williams and Roussenov (2014) show that the return flow is indeed baroclinic on an interannual timescale. Our results support this finding but it is not the main focus of our research. However, it is important to understand the geostrophic flow of the AMOC and disentangle different contributions of the meridional velocity field in order analyze whether the level of no motion is a proxy for northward transport. The level of no motion is a proxy for the total maximum overturning streamfunction despite the Ekman cells because the surface Ekman flux is compensated baroclinically above the level of no motion. Also commented and a matter of writing, in Fig. 8 we assume that the vertical velocity shear is constant because we would like to

show how much of the changes in volume transport between the wind experiments is attributed to the displacement of the level of no motion only. We make better connections, which is outlined in the response to comment II.

**IV) REFEREE 1**

*In general, almost every sentence is difficult to read. The authors add words to the sentences that do not give any information, and they omit words that would specify what they are talking about. Often the connection between one sentence and the next is not clear. Below, I give some examples of difficult paragraphs. Unfortunately, many paragraphs have similar problems. It would help if paragraphs were not so long. For instance in the 2nd paragraph of the Intro, "The depth scale itself is determined..." can start a new paragraph, "Different assumptions like..." can start a new paragraph, and "The present study addresses" can start a new paragraph. Similarly, paragraph 3 can be divided into 3 or 4 paragraphs, as can many other paragraphs throughout the paper. Ideally each paragraph gives one main idea, otherwise its easy for the reader to get lost in overlong blocks of text.4*

As stated above, we spent much effort into the writing style of the new paper. As stated above, the author's writing style is to write in a condensed way but having transitions between the sentences in order to facilitate comprehension. We meet the comments on the writing style made by the referee and now write in a more simple way and add definitions such as defining *wind stress curl* or *nonlocal and local*. Focusing more on the wind forcing dependence of the AMOC now, we state more clearly the questions that are being asked and why we need the experiments or wind patterns. Finally, the author's style is to have paragraphs that describe qualitatively proper sections. We reviewed the sections in order to ensure that the reader can follow the arguments more easily. We tried to improve the writing throughout the paper. We do not comment on the writing examples given by the referee but meet them during our revision.

We do not list the changes made but we improved the writing throughout the manuscript. We make better connections and transitions.

All minor comments or suggestions on the writing style are met.

Small issues:

*Sec 1 para 3: paragraph emphasizes lack of work on effects of northern hemisphere wind on overturning, but Klinger et al. (2003) and (2004) both looked a effect of Westerlies in both hemispheres.*

There is a lack in literature regarding the northern hemisphere downwelling region where Ekman pumping takes place. Your propositions are very valuable but would distract from the major flow of the paper.

**Literature**

Gutjahr, O., Putrasahan1, D., Lohmann, K., Jungclaus, J. H., von Storch, J.-S., Brüggemann, N., Haak, H., and Stössel, A.: The Max Planck Institute Earth System Model (MPI-ESM1.2) for the High-Resolution Model Intercomparison Project (HighResMIP), Geoscientific Model Development, 12, 3241–3281, https://doi.org/10.5194/gmd-12-3241-2019, 2019.

Levang, S. J. and Schmitt, R. W.: What Causes the AMOC to Weaken in CMIP5?, Journal of Climate, 33, 1535–1545, https://doi.org/10.1175/JCLI-D-19-0547.1, 2019.

McCreary, J. P. and Lu, P.: Interaction between the Subtropical and Equatorial Ocean Circulations: The Subtropical Cell, Journal of Physical Oceanography, 24, 466–497, https://doi.org/10.1175/1520-0485(1994)024<0466:IBTSAE>2.0.CO;2, 1994

Williams, R. G. and Roussenov, V.: Decadal Evolution of Ocean Thermal Anomalies in the North Atlantic: The Effects of Ekman, Overturning, and Horizontal Transport, Journal of Climate, 27, 698–719, https://doi.org/10.1175/JCLI-D-12-00234.1, 2014.

---

## Author Comment (AC4)

**Rohrschneider et al. 2021, response letter**

**The depth scales of the AMOC on a decadal timescale**

With the authors' response we re-initiate the publication process and soon submission of the revised manuscript. We are grateful for the many comments which have been posted by the anonymous referees. In the following we outline the future changes on the present manuscript. Our study is based on two sets of experiments only but we believe it will be a major contribution to the scientific community because the local wind forcing dependence of the AMOC has not been explored yet.

We changed the manuscript substantially. We focus on the wind experiments only and neglect the global warming experiment. The content of the paper is now well summarized by the abstract.

We use wind sensitivity experiments to understand the wind forcing dependencies of the level of no motion as the depth of maximum overturning and the e-folding pycnocline scale as well as their relationship to northward transport of the mid-depth Atlantic meridional overturning circulation (AMOC). In contrast to previous studies, we investigate the interplay of nonlocal and local wind effects on a decadal timescale. We use 30-year simulations with a high-resolution ocean general circulation model (OGCM) which is an eddy-resolving version of the Max Planck Institute Ocean Model (MPIOM). Our findings deviate from the common perspective that the AMOC is a nonlocal phenomenon only, because northward transport and its depth scales depend on both nonlocal Southern Ocean wind effects and local wind effects in the northern hemisphere downwelling region where Ekman pumping takes place. Southern Ocean wind forcing predominantly determines the magnitude of the pycnocline scale throughout the basin, whereas northern hemisphere winds additionally influence the level of no motion locally. In that respect, the level of no motion is a better proxy for northward transport and mid-depth velocity profiles than the pycnocline scale, since the wind forcing dependencies of the level of no motion and maximum overturning are equal. The changes in maximum overturning with wind forcing are explained by the changes in the level of no motion only. This is because wind-driven Ekman compensation is baroclinic and occurs above the level of no motion, and the internal vertical velocity shear that is not influenced by the external Ekman cells stays approximately constant. The analysis of the wind experiments suggests a hemisphere-dependent scaling of the strength of AMOC. We put forward the idea that the ability of numerical models to capture the spatial and temporal variations of the level of no motion is crucial to reproduce the mid-depth cell in an appropriate way both quantitatively and dynamically. (line 1-17)

We answer explicitly the following research questions.

This paper presents an analysis of wind sensitivity experiments in order to provide insight into the wind forcing dependence of the inter-hemispheric circulation by understanding the behavior of the depth scale(s) of the AMOC. (line 28-30)
…
Understanding the wind forcing dependence of the AMOC by understanding its depth scales makes the underlying research question twofold, in the sense that we discuss the wind forcing dependence of the AMOC using the depth scales and we discuss whether the depth scales are proxies for northward transport to understand the wind forcing dependence. We hypothesize that the level of no motion is a proxy for northward transport in the inter-hemispheric cell because the background vertical velocity shear of the meridional velocity may stay constant under changing wind forcing. (line 62-67)

…
We focus on the inter-hemispheric region 30S-30N and analyze the interplay of nonlocal wind forcing over the Southern Ocean and local wind forcing in the northern hemisphere downwelling region where Ekman pumping takes place. (line 70-71)
….
We address the question how changes in both nonlocal and local wind forcing influence the AMOC. We hypothesize that the influence of northern hemisphere winds on the AMOC is substantial. (line 85-86)

We changed the discussion accordingly and discuss the wind forcing dependence of the AMOC, the mechanism of the wind forcing dependence, and the depth scales as proxies for meridional

flow. Theses topics are not limited to the discussion but the major focus of the present manuscript. The paper is much more tailored regarding the focus of the present manuscript. In this connection, we did rewrite the introduction, say why our high-resolution simulation is necessary, elaborate on the mechanism of the wind forcing dependence with a new figure, inter alia.

We improved the writing considerably throughout the manuscript. In this response letter, we cannot state all changes made, but we made changes in every section in order to improve the flow of the paper.

Please review both response letters as they are entangled due to the sheer amount of the changes made.

**I)-V) : major comments**

**VI) : minor comments**

*:comment*

:response

**V) REFEREE 2**

*But my main difficulty here: which of these is the main focus of the paper, and what is the take-away message? About 3/4 of the paper focuses on the wind experiments, and if the title were changed to reflect this topic, the paper would very much read like this was the primary subject. But the paper also analyzes the global warming experiment, which (given the ms. title) suggests that this study is meant more as a contribution about AMOC depth scaling and theory (?) So which is it?*

A main difficulty stated by the referee is the main focus of the paper. The aim of the study is to understand the wind forcing dependence of the AMOC by understanding its depth scales. That is to say, it is twofold in the sense that we discuss the wind forcing dependence of the AMOC and we discuss whether the depth scales are proxies for northward flow to understand the wind forcing dependence. Hitherto, the main focus on the wind forcing dependence of the AMOC suffered in some parts of the wind experiments and in the global warming experiment. We therefore strengthened the main focus on the nonlocal (Southern Ocean) and local (northern

hemisphere downwelling region) wind forcing dependence throughout the present paper.

Besides adding more context in each chapter, we do so now by rewriting section (4) on the wind experiments (the relationship between the depth scales and meridional velocity profiles), as well as *Introduction* (1) and *Discussion* (5). Considering section (4), we now focus on the mechanism on the wind forcing dependence of the AMOC and refer to the experimental setup of the present study. Deep velocity shear below the advective depth remains nearly constant, whereas the velocities above the advective depth are altered, which together changes the level of no motion. In summary, section 4 provides the opportunity to understand the wind forcing dependence of the AMOC by analyzing the relationship between the depth scales and meridional velocity profiles in more detail, and disentangle different contributions. We now generate some insights on the maximum overturning streamfunction in the case that the vertical velocity shear is constant with wind forcing, using the 1X reference experiment to compute the constant vertical velocity shear. This maximum overturning streamfunction is the maximum northward transport that arises from the changes in the level of no motion only, and we find that it explains the changes with wind forcing in very large degree by vertical integration. Please refer to the response of comment III.

We now focus on the wind experiments only. We neglect the warming experiment because it is not the main focus of the paper. We now discuss wind forcing dependence and global warming in *Discussion* (5). In a future study, we would like to additionally analyze whether local Ekman pumping in the northern hemisphere downwelling region substantially influences the adjustment of the AMOC to warming.

The discussion mirrors the content of the paper, which has been changed accordingly throughout the text. We propose to change *Discussion* (6) in the following way.

line 375-418

In line with the current understanding of the Atlantic circulation, Southern Ocean winds boost the strength of the AMOC and change density stratification throughout the basin \citep[e.g.][]{vallis2000,klinger2003,klinger2004,klinger2009}. Northern hemisphere winds over the downwelling region additionally influence the meridional flow and density stratification locally, which is commonly ignored in the scientific literature on the AMOC. The present study is based on simulations with an eddy-resolving OGCM on a decadal timescale rather than a fully equilibrated experiment. We find a robust adjustment of the AMOC and density field, which demonstrates the realization of major adjustments due to wave propagation, and the 30-year simulations are long enough to analyze the wind forcing dependencies of the depth scales and northward transport. The wind forcing-dependence of the AMOC is reflected by the wind experiments.

The findings of the present study support the pycnocline model described in \citet{gnanadesikan1999} in the sense that Southern Ocean wind forcing deepens the pycnocline scale and the level of no motion and strengthens the AMOC. However, local wind forcing over the northern hemisphere downwelling region additionally influences the level of no motion and northward transport locally. In that respect, the level of no motion is more appropriate to scale northward transport than the pycnocline scale. By artificial

modification of density gradients in OGCM experiments, \citet{griesel2006} and \citet{deboer2010} indicate that the pycnocline scale does not scale northward transport at all. By contrast, we provide insight on the scaling behavior of the depth scales from a conceptual point of view, and the pycnocline scale fails to scale northward transport in the northern hemisphere.
\

Wind stress curl variations at the surface translate into changes in the AMOC. The changes of the AMOC with changing wind forcing in the inter-hemispheric region are explained by the changes in the level of no motion. The internal velocity shear that is not influenced by the external Ekman cells remains constant on the timescale considered here. In contrast to what is stated in \citet{cabanes2008} who analyze interannual variability, the forcing imposed by the wind stress curl at the surface does not substantially change the vertical shear but the reference depth of the AMOC shear component. Our findings also deviate from \citet{levermann2010} who evaluate the pycnocline model using a model of intermediate complexity. They analyze equilibrated experiments which reproduce the response to Southern Ocean wind forcing and focus on meridional density gradients instead of zonal density gradients to represent vertical velocity shear. Using meridional density gradients instead of zonal density gradients is based on the assumption that these gradients are proportional and have the same order of magnitude, and zonal and meridional velocities compare well with one another. According to their findings, both the pycnocline scale and meridional density gradients vary, while according to our study the internal velocity shear remains fixed. We speculate that our high-resolution simulation better simulates velocity shear.

The displacement of the level of no motion in the MPIOM wind experiments approximates the conditions in the interior with the Ekman cells mainly cancelled out. Comparing the wind experiments, the ocean response at the upper levels is much more complex than the response at the deeper levels, which is mostly related to the baroclincity of the interior return flow of the surface Ekman flux. However, integrating vertically, the changes that are associated with the level of no motion give approximately the changes in the total maximum overturning streamfunction with changing wind forcing. As a general contribution and supporting the theoretical considerations made in \citet{mccreary1994}, our findings give baroclinic Ekman compensation which has been demonstrated in an idealized way by \citet{williams2014}. Baroclinic Ekman compensation may depend sensitively on the resolution of an OGCM.

The wind forcing dependence of the AMOC suggests that the temporal adjustment of the AMOC to global warming is not independent of location. Both nonlocal Sothern Ocean wind forcing and local wind forcing in the northern hemisphere downwelling region are likely to influence the adjustment of the level of no motion and northward transport in the inter-hemispheric region.

We are going to keep the experimental strategy which is consistent with the necessary changes.

**VI) REFEREE 2**

*As one example of what seemed underwhelming in regard to building on AMOC theory, while the Bryan 1987 scaling is cited and its general approach explained, the ms. does not explain how this has been used to predict scaling of the AMOC to different parameters, the most explored being diapycnal mixing, but also density gradients. Might this be relevant in the 4XCO2 expt? And although Levang and Schmitt is cited, there is little discussion of how new findings here might relate to their study (note, not sure I understood the sentence l. 421). I would think the large spread of AMOC changes in CMIP-class models might be excellent motivation for the 4xCO2 experiment, If the authors wished to explore this further.*

We would have added considerations on advective-diffusive balance (Bryan 1987) in order to add some explanations on the pycnocline scale in the global warming experiment. However, we neglect the global warming experiment in the new manuscript. The connection to Levang and Schmitt (2019) is simple. In the northern hemisphere downwelling region, positive salinity-induced changes in density counterbalance negative temperature-induced changes in density. The latter causes a weakening of the AMOC because the geostrophic shear component of the AMOC is altered. The anomalies in density (salinity and temperature) are advected from the surface to

deeper levels due to the forcing imposed by the wind stress curl at the surface. In general, there is less spread in the wind stress curl at the surface among CMIP6 models but the translation of the forcing into density and pressure anomalies at the advective depth remains a major source of uncertainty. The authors look forward to explore the influence of local Ekman pumping on the AMOC in CMIP6 global warming experiments in a subsequent study.

*Motivation for model choices was lacking. If this work was intended as a more conceptually-oriented contribution, could this be accomplished using a coarse, idealized setup? Or maybe, both a (cheap) coarse, idealized setup could be contrasted with the realistic run? While there is some hint in the ms. that the eddying capability is important in particular for accurate wave-propagation (given the decadal adjustment timescale), this is not clear to me. Or, why not run a coarse model to equilibrium? Is the decadal adjustment an element of the story (i.e. as implied in title) ? I'm not saying this study needs to be redone with a different setup, just that there is scant justification for the model setup used.*

We make considerations on the model choice and experimental strategy above (I).

*On figures, many lines labelled as "black" were more gray to me, and "opaque" vs. "transparent" were better described as red vs. pink, for example. Is dotted blue line missing in 8a?*

The dotted blue line and the dotted red line overlap, since the level of no motion between the 2XSH and 2X experiments is approximately equal south of the equator.

*l. 40 discusses "diapycnal upwelling in the tropics" after mentioning the Gnanadesikan (1999) model. Although perhaps a bit beyond the scope of this paper, I might argue Gnanadesikan is a single basin model that explicitly assumes the adv-diff balance in its overturning hemisphere, whereas the model here is global and one might assume the important advective-diffusive balance (justifying an e-folding pynocline scaling) might be occurring in the (larger) Pacific basin. Again, if the main focus is as a conceptual AMOC contribution, it is a bit disappointing to not even comment on other possible relevant issues such as this, nor advance the science with new or revised conceptual models, nor use new results to go back and comment more extensively on conceptual understanding in the literature.*

We switched the focus of the paper and narrowed the research questions. We do not raise expectations which cannot be met. Commenting other relevant components such as the global

pycnocline is beyond the scope of the paper now.

*It would seem more could be explored about the relationship between zonal and meridional density gradients. Of course, the pycnocline scaling says nothing directly about zonal density gradients, in contrast with level of no motion (albeit somewhat indirectly).*

We strengthened the focus on the wind forcing dependence of the AMOC. Therefore, it is nearly impossible to include an analysis on the relationship between zonal and meridional density contrasts. We focus on the role of zonal density contrasts and vertical velocity shear with respect to the changes that are related to the level of no motion.

*l. 116 mentions "monthly climatology of reanalysis wind stress is doubled"; by this I presume the reanalysis wind is available every six hours or thereabouts, and the six-hourly variability is preserved but the monthly mean wind is doubled? Or please explain. Does high-frequency forcing play any role?*

We only double the monthly mean climatology of the wind stress curl while the anomalies are unchanged. We do not consider high-frequency variability of the AMOC.

*l. 153 "surface buoyancy fluxes change continuously": this could be said about any model with a seasonal cycle or interannual forcing. I think what is meant is that the 4xCO2 experiment adjusts slowly to the step change, with the surface forcing changing as a function of ocean-atmosphere state in the coupled setup. In contrast the wind experiments are not coupled, and the adjustment is assumed to occur on a decadal time scale, simplifying the analysis to a comparison of 1991-2010 mean states*

The warming experiment is based on step function forcing of the radiative forcing or the atmospheric CO2 concentration. We would rewrite the statement that the surface buoyancy fluxes change continuously in order to explicitly state that they change constant in sign as a function of time. The transient nature of the 100-year global warming experiment would have made it possible to compare it with the mean states of the wind experiments on a decadal timescale.

*l. 280 by "inter-hemispheric regions" I presume the authors are referring to 30S-30N?*

We consider the region away from the lateral margins; that is, 30S-30N.

*l. 452 didn't follow sentence*

The changes in the level of no motion between the wind experiments explain a large fraction of the changes in meridional velocities. Near the surface, however, the signal that arises from the interior return flow of the surface Ekman flux overcomes the signal that arises from the displacement of the level of no motion

**Literature**

Gutjahr, O., Putrasahan1, D., Lohmann, K., Jungclaus, J. H., von Storch, J.-S., Brüggemann, N., Haak, H., and Stössel, A.: The Max Planck Institute Earth System Model (MPI-ESM1.2) for the High-Resolution Model Intercomparison Project (HighResMIP), Geoscientific Model Development, 12, 3241–3281, https://doi.org/10.5194/gmd-12-3241-2019, 2019.

Levang, S. J. and Schmitt, R. W.: What Causes the AMOC to Weaken in CMIP5?, Journal of Climate, 33, 1535–1545, https://doi.org/10.1175/JCLI-D-19-0547.1, 2019.

McCreary, J. P. and Lu, P.: Interaction between the Subtropical and Equatorial Ocean Circulations: The Subtropical Cell, Journal of Physical Oceanography, 24, 466–497, https://doi.org/10.1175/1520-0485(1994)024<0466:IBTSAE>2.0.CO;2, 1994

Williams, R. G. and Roussenov, V.: Decadal Evolution of Ocean Thermal Anomalies in the North Atlantic: The Effects of Ekman, Overturning, and Horizontal Transport, Journal of Climate, 27, 698–719, https://doi.org/10.1175/JCLI-D-12-00234.1, 2014.

---

## Author Response (AR2)

REVIEW 1

Nonlocal and local wind forcing dependence of the Altantic meridional overturning circulation and its depth scale

COMMENT
REPLY
CITITATION

Dear reviewer, first of all we would like to thank you for your effort. We were happy to address the major comments in an extended way. We are afraid that we neglected a wide range of the minor comments because we believe these comments are driven by personal belief with respect to an own paper. We hope the paper is publishable according to your opinion which is important to us.

MAJOR COMMENTS

Overview

Revised manuscript is much more coherent, though still not easy to read. It argues that in a global ocean model, the Southern Ocean wind stress controls the Atlantic pycnocline depth, but that the Northern Hemisphere wind stress also influences the thickness and magnitude of the upper limb of AMOC in the northern hemisphere. I recommend one more set of modifications before publication, but I now believe it should be eventually publishable after one more revision.

Main Comments

A. Time Dependence. One essential revision is the need to characterize the time-dependence of their results. While extending the model runs may be prohibitively expensive due to the resolution, the paper should at least talk about the time-dependence for the 30 years of the run. Is there any evidence from the behavior of the overturning or stratification that the 20-year averages taken here would be similar if the next 20 years were used? Is there any evidence that the model is converging so that results may be similar if the run was extended another few centuries? While the paper explicitly says it is only talking about a particular time period, the theoretical framework of the paper is based on steady-state behavior, so its incomplete to not comment further on whether the results are relevant to the steady-state. And if the results are to be relevant to the transient behavior, then the time evolution has to be discussed.

We added a proper section on the robustness of the results (please review the content below)  line 407-456

B. Longer low-resolution runs. The paper would be better if an additional set of long runs were done with a non-eddy-resolving grid. This would tell us if the eddy resolution is important for getting the correct sensitivity to wind, and would give further insight into whether the results are indicative of long-term means. This is plausibly beyond the scope of the current paper, so I won't insist on it, but at least a second paper to check this would be worth considering.

We added a proper section on the robustness of the results (please review the content below)  line 407-456

**\section{Robustness of the wind forcing dependence}**
In this section we would like to elaborate on the robustness of the results
considering the wind forcing dependence of the AMOC. The question arises whether
the wind forcing dependence of the AMOC found in the short-term integrations of
TP6ML80 (1980-2010) is robust.  In the study we use the time window (1991-2010) in
order to allow for major adjustments at an initial stage. We state that the wind
forcing dependence found in the time window would reflect a quasi-steady response.
This is a strong assumption given that it is actually a transient response within a
short integration time. The adjustment in the density field (Fig. 3) support the
perspective that major adjustments in ocean dynamics to forcing are realized.
However, the wind forcing dependence of the AMOC may still be time-dependent, and
low-resolution model outcome may differ from high-resolution model outcome.
\newline

We first show the full time series (1980-2010) of maximum overturning and the level
of no motion in TP6ML80 (30S-10S,10N-30N) (Fig. 10 a,b,c,d). There is a strong
adjustment and time-dependence in both variables at an initial stage on a decadal
time scale. During the course of the study we have neglected this initial
adjustment by focusing on the time window 1991-2010 only. After the initial
adjustment on a decadal timescale (1980-1990), the wind forcing dependence of
maximum overturning and the level of no motion is robust. Nevertheless, there are
oscillations at low frequency which put into question whether the wind forcing
dependence of the AMOC found in the short-term integration of TP6ML80 is quasi-
steady. We cannot investigate the steady response of the AMOC in TP6ML80 due to the
high computational costs. The temporal changes in the level of no motion, however,
coincide with the temporal changes in maximum overturning in the sense that the
vertical velocity shear of the meridional velocity stays approximately constant
over time.
\newline

\begin{figure*}
**\includegraphics**[width=0.9\textwidth]{time_image}
\caption{The TP6ML80 time evolution (30S-10S,10N-30N) of (a,b) maximum overturning
and (c,d) the level of no motion after the forcing is switched on. The MPIESM1.2-LR
time evolution (30S-10S,10N-30N)of (e,f) maximum overturning and (g,h) the level of
no motion after the forcing is switched on.}
**\label{fig:41}**
\end{figure*}

\begin{figure*}
**\includegraphics**[width=0.9\textwidth]{warming_image}
\caption{The time evolution (30S-10S,10N-30N) of (a,c) maximum overturning and
(b,d) the level of no motion after the forcing is switched on in the global warming
experiments with altered surface wind stress using MPIES1.2-LR. Atmospheric CO$_2$
is quadrupled.}
**\label{fig:41}**
\end{figure*}

As a next step, we use a AGCM-OGCM coupled low-resolution model to simulate the
wind forcing dependence in a low-resolution counterpart and on a longer timescale
(50 yr). The coupled model better simulates the salinity balance in the OGCM to
which ocean dynamics are sensitive. The coupled model is MPIESM1.2-LR, with the
low-resolution configuration of MPIOM being the OGCM component. The ocean model
(GR15L40) has a horizontal resolution of 1.5 degrees and 40 vertical levels only.
We have a set of four experiments: the 2X experiment in which the zonal and
meridional surface wind stress is doubled throughout the hemispheres; the 2XSH
experiment in which the wind stress is doubled over the Southern Ocean only; the 1X
experiment which is forced under no changes; and the 0.5X experiment in which the

zonal and meridional wind stress is halved. We only change the ocean wind stress
factor that multiplies the surface wind stress in the coupled model because I am
interested in the OGCM dynamics only. It is an online multiplication of each wind
stress value at each timestep.
\newline

We find that in the 50 years integrations of the low-resolution model the response
is apparently quasi-steady on this timescale (Fig. 10 e,f,g,h). On longer
timescales, internal, low-frequent variability may take place. We find that the
wind forcing dependence of maximum overturning is similar to TP6ML80 and robust.
However, there are major deviations in the level of no motion which does not
reflect the wind forcing dependence in the high-resolution model outcome. The
general finding that the level of no motion deepens with stronger wind forcing is
confirmed, but the details between the 1X and 2XSH experiments are not well
simulated. This may be due to model drift in the coupled model, or low-frequency
oscillations, or the low vertical model resolution. The level of no motion is
sensitive to small variations in the velocity field which may still adjust and
oscillate. It seems that the nonlocal wind forcing dependence of the AMOC is less
strong and the local wind forcing dependence is much stronger. The vertical
velocity shear of the meridional velocity is not constant.
\newline

Disappointed from the finding that the level of no motion may not be well
represented in the low-resolution MPIOM configuration, we looked for an alternative
way to make sure that the wind forcing dependence of the AMOC is robust. We
computed the wind forcing dependence of the AMOC in 100-year global warming
experiments with altered surface wind stress, using also MPIESM1.2-LR. We
quadrupled atmospheric CO$_2$ and applied the wind stress factor during the forward
integration. We initialized with the control experiments with altered surface wind
stress at year 30, after having explored that the initialization plays a minor role
for the evolution of the AMOC in the global warming experiments. We believe the
system is more strongly forced so that the forced underlying dynamics overcome
internal oscillations and model drift. Fig. 11 shows the wind forcing dependence
(30S-10S,10N-30N) of maximum overturning and the level of no motion in the global
warming experiments with altered surface wind stress. Now the wind forcing
dependence of maximum overturning and the level of no motion is the same as in the
wind sensitivity experiments with TP6ML80.

C. More on Dynamics. I simple way of looking at the effect of Southern Ocean wind stress is that it's
Ekman transport pushes water northward, and the resulting current joins the upper limb of the AMOC
and returns with NADW. Shouldn't the northern hemisphere wind do the opposite? Isn't that what
happens in previous studies? Ekman transport from strengthened westerlies goes southward, which
would weaken the upper limb rather than strengthening it. Similarly, that same stronger wind in
subpolar gyre would increase upwelling, which might counteract the downwelling associated with deep
water formation. It would be helpful if the paper addressed this point. In addition, a key result is that
we can think of a change in NH wind moving the depth of maximum streamfunction vertically, which
moves a fixed vertical velocity shear vertically, which then determines the change in the overturning.
But why should each of these facts (change in depth, constancy of velocity shear) be true? Answers to
these questions would enhance the paper though they are not required for publication.  line 498-512

We added a section on dynamical components considering our study to
the end of the discussion section. Note that the focus of the paper
is the new way how to decompose the AMOC and explain the wind forcing

dependence. AMOC components away from the inter-hemispheric region play a minor role. Please review the content below.

We would like to use this discussion section to refer briefly to the wind forcing dependence of the AMOC in terms of dynamic components. An outcome of our experimental study is that northward overturning is well approximated by the level of no motion which reflects the wind forcing dependence of the AMOC. We demonstrate that, using the level of no motion, the  flow can be subdivided into internal flow and external flow, because the external baroclinic Ekman cells that are directly forced by the surface winds cancel out by vertical integration. Our findings support baroclinic Ekman compensation which makes the level of no motion a proxy for northward overturning. That is to say, meridional Ekman transport in the southern hemisphere as well as in the northern hemisphere do not change the relationship between overturning and its depth. Thus, it does not change the wind forcing dependence of the AMOC because the surface Ekman flux is compensated above the level of no motion. Ekman pumping in the southern hemisphere and in the northern hemisphere do change the relationship between overturning and its depth. The explanation for the changes in maximum overturning and the level of no motion differs between the southern hemisphere and the northern hemisphere. In the southern hemisphere north of the ACC Ekman pumping displaces isopycnals downward that span the basin meridionally. In the northern hemisphere the increase in transport and depth can be explained by continuity and isopycnals are displaced downward only locally. The wind forced change in Ekman puming gives a new advective balance. It forces the flow thus horizontally upstream, and a new dynamical balance establishes downstream. We speculate that in this way maximum overturning and the level of no motion are altered.
\newline

MINOR COMMENTS:

1. Improve Table 1. Make table less wordy and leave out information common to all 3 experiments, so that table looks something like this:

   Abbreviation Name Description
   1X Reference Observed wind stress
   2XSH Double Southern Wind Double wind stress south of 30S
   2X Double Wind Double wind stress at all latitudes

We do not believe that is necessary to change Table 1 and prefer it the way it is.

2. Improve Table 2. Separate and organize variables into groups, make separate "parameter name" and "parameter definition" columns. At the authors' discretion, I suggest using lower case and upper case symbols to differentiate between variables and parameters, using a capital Z or H rather than lower-case η for depths, referring to "level of no motion" as "Streamfunction depth" or "upper-limb depth", and referring to "pycnocline scale" as "pycnocline depth" (because it refers to a measurement of model behavior, unlike the advective depth scale which is an estimate calculated from forcing parameters." I don't see the need for defining derivatives of ψ in the table because (for instance) $\partial\psi/\partial z$ obviously means the "vertical derivative of the overturning Streamfunction".

We believe it is straightforward to capture the content of the Table.
We have our own definitions which we use throughout the paper in a
consistent way. It is technical.

3. Clarify model spin-up (Sec 2.1). Paper says "we focus on the time-window 1991 to year 2010". How long before that (if at all) was the model spun up? How close to steady state was it at this point? What does "change the monthly-mean climatology of the surface wind stress only" mean? Does run 1X use daily wind stress, and other runs use daily wind stress multiplied by a factor, or is there some more complicated procedure involving taking monthly means? Or are monthly mean wind stress used for all runs?

Thank you, we did clarify this in the paper.

4. "Mid-depth" is confusing. In Abstract and elsewhere, I suggest replacing "mid-depth AMOC" with "upper limb of the upper AMOC cell" or "upper limb of the North Atlantic Deep Water (NADW) cell of AMOC".

Done

5. Density Difference Figure. I don't understand what is gained by looking at density differences in Fig 3ab. Since the reason for examining density is connected to measurements of isopycnals depth, why not just look at isopycnals? Also, rather than using the normalized density (black curves in Fig 3ab), it would be better to use a measure that is closer to the one used to calculate integral depth scale
$r(y,z)=(\int_z^0 (\rho-\rho_r )dz)/(\int_{(z_T)}^0 (\rho-\rho_r )dz)$
Which by definition gives r=0 at the surface and r=1 at z=z_T. Can then show separate panels for r contours for each experiment.

We do not agree with you. It is straightforward to capture the
difference in zonal-mean density or stratification in our study.
Furthermore, we define the term stratification and use it
consistently throughout the paper.

6. Focus on latitude-band averages for depth. Figure 6 shows averages over latitudes 10-30o in both hemispheres. Extend this to several quantities. Instead of Fig 3c and Fig 4 showing latitude dependence, just show averages for each of the 3 depth scales in each hemisphere. Each panel would contain depths (y axis of panel) for 3 runs (x axis of panel) for northern (upward-pointing triangle) and southern hemisphere (downward-pointing triangle) for a single quantity (streamfunction depth, pycnocline depth, advective depth scale). This would emphasize how each quantity depends on the wind, rather than current version which emphasizes complicated latitude dependence which text does not comment on much. Also Fig. 4 currently is very busy with 6 different curves and one has to concentrate to see the point about the wind different wind dependence in NH and SH. The latitude dependence of $\eta_w$ is not an appropriate value to plot, since $\eta_w$ is a scale quantity representing the pycnocline depth for a given gyre, not the detailed geographical variation of pycnocline depth within the gyre.
We do not agree. We think showing a latitudinal dependence of the
quantities is the most tangible way to illustrate the wind forcing
dependence of the AMOC.

7. Why is advective depth scale included? Currently fig 3. Plots $\eta_w$ (y) with different values of g'. The values seem arbitrary, and I don't understand why these alternate calculations are graphed. The only significance I can see of $\eta_w$ is that it depends on $\sqrt{\tau}$. Therefore, maybe just compare $\eta_\rho$ variations to $\sqrt{\tau}$ in the plot I suggest in (6) above.

```
We do not agree. The advective depth scale is directly related to
local Ekman pumping or the wind stress curl at the surface.
Furthermore, different g' s show the possible parameter space.
```

8. What is the significance of the geostrophic transport? Below the Ekman layer (top 50 m or less?), shouldn't velocity be geostrophic? Or does nonlinearity from the eddies add an important term? The max geostrophic streamfunction shown in Fig 5 is some kind of perturbation due to the Ekman transport? How is it relevant to the discussion of the overturning? If it isn't, why is it discussed?

```
We added explanatory statements to this section.
```

9. Maybe separate transport and depth data. Since I think the depths should be plotted as a function of experiment, maybe the NH and SH transports should be plotted that way as well rather than plotted against depth as in Fig 6. Then again, the plot does do a good job showing that volume transport varies with $\eta_\psi$, so I wouldn't object to keeping it anyway. The dashed lines are distracting though; if graph kept as-is, eliminate them and perhaps use a more distinct symbol if geostrophic data is retained in revision – perhaps open symbol instead of lighter symbol. The transport estimated from shear and $\eta_\psi$, currently shown as a function of latitude in Fig 9, could also be included in the figure (10-30o average for each hemisphere as a function of run).

```
We do not agree. We think the way the figures are displaced is the
best way to understand the content of the paper.
```

DONE

11. Abstract Clarity. The Abstract is okay as written, but has a number of awkward elements. Here I list those elements and give an alternative text for the first 2/3 of the Abstract. The authors can use all, part, or none of the alternative text at their discretion.
"wind forcing dependencies" is a little vague
"level of no motion as the depth of maximum overturning" is trying to say that the 1st phrase = 2nd phrase, but readers may be confused by "as the"
"interplay of nonlocal and local" also kind of vague – at this point reader still doesn't really know what abstract is talking about
"downwelling region where Ekman pumping takes place" Actually the wind is changed over entire hemisphere, so not clear that it's the Ekman pumping location that is key
In my rewrite, I try to give the reader a bit more context first, and to describe the issues and experiments in a more concrete way.

```
We do not agree. The abstract is written in a technical way, in a way
that corresponds to the content of the paper. It is technical. We do
not believe that is necessary to show further content and context.
```

REVIEW 2

Nonlocal and local wind forcing dependence of the Altantic meridional overturning circulation and its depth scale

COMMENT
REPLY
CITITATION

Dear reviewer, first of all we would like to thank you for your effort. We were happy to address the major and minor comments. We hope the paper is publishable according to your opinion which is important to us.

I appreciate the focus now on the depth scale and wind experiments (removing the 4xCO2 part), and while this is an improvement, some problems remain.

Starting with the introduction and motivation, in my opinion the authors raise two important questions that are often overlooked or not considered especially relevant, specifically 1) regarding the relevant AMOC depth scale and 2) regarding local vs. non-local winds. The content here is good, and appropriate references are ticked off, but there's a remaining problem: (given that 1&2 ostensibly seem so different) which of these is the main question and which is used in experimental analysis/support? From the title, the depth scale seems the main question. But reading the abstract and the intro, the wind question seems primary. Which was the question the authors first considered? I think one could take 1 as the main question as explored via 2's experiments, or take 2 as the main question and hypothesize 1 is the relevant diagnostic to consider/analyze. My point here being that the presentation of the intro comes across as a bit of a jumble, and improvement therein might help with my remaining problems in the manuscript. These two questions are obviously not totally unrelated; some better transitions, motivations, connections in the text etc would help.

We changed the introduction accordingly. We rearranged the structure of the introduction and added explanatory statements. Basically, we analyze the wind forcing dependence of the AMOC and explain the wind forcing dependence of the AMOC by the relationship between overturning and its depth. Considering the introduction, we now first discuss the wind forcing dependence of the AMOC and then explain why we analyze depth scaling and thermal wind.

The structure of the introduction now corresponds to the structure of the paper. line 53-71

The research of the present study is inherently about depth scaling that reflects the wind forcing dependence of the AMOC, because we understand the wind forcing dependence of the AMOC by the behavior of its depth scale. Oceanographers use theoretical scaling relationships to provide conceptual understanding and to estimate the strength of the AMOC in response to different forcings.
…
Understanding the wind forcing dependence of the AMOC by understanding its depth scales makes the underlying research question twofold, in the sense that we discuss the wind forcing dependence of the AMOC using the depth scales and we discuss whether the depth scales are proxies for northward transport to understand the wind

forcing dependence. The latter question is implicit in the sense that we need to
answer this questions in order to explain the wind forcing dependence of the AMOC.
We hypothesize that the level of no motion is a proxy for northward transport in
the inter-hemispheric cell because the background velocity shear of the meridional
velocity may stay constant under changing wind forcing. In this connection, the
study is based on different ways or definitions which describe meridional flow in
order to analyze how the changes in wind forcing are translated into the changes in
the AMOC. We demonstrate that, using the level of no motion, the flow can be
subdivided into internal flow and external flow, because the wind-forced Ekman
cells, which give the Ekman transport and its compensation, are found to be
baroclinic and cancel out by vertical integration above the level of no motion. The
internal flow is directly related to the AMOC wind forcing dependence.
\newline

I found going through the results was difficult; it took multiple read-throughs and a fair amount of
effort. Below I identify some specific points requiring clarification, but in the bigger picture, main take-
aways need to be better identified; in some places I did not follow what questions or points the analysis
was attempting to address.

We added explanatory statements.

That being said, a major point seems that eta_rho doesn't change between 2xSH and 2X whereas
eta_psi (and psi itself) do. There is some follow-up on what this result suggests in terms of previous
work (in section 4), but fairly cursory, and if this is indeed a significant result, more could be said in
this regard, even if previous work seems a bit tangential in experiments and conclusions.

We explain it now in an explicit way.  line 323-334

At this point, we would like to summarize why it is important to distinguish
between the pycnocline scale and the level of no motion considering the scaling of
maximum overturning. In this way, we avoid a tangential analysis. The pycnocline
scale does not scale maximum overturning, whereas the level of no motion scales
maximum overturning. In addition, the pycnocline scale cannot capture the details
of stratification, and at the same time we cannot capture the wind forcing
dependence of the AMOC when only knowing how density unfolds vertically. The
pycnocline scale is commonly taken as appropriate depth scale in current literature
but actually it does not reflect the wind forcing dependence of the AMOC. The level
of no motion does reflect the wind forcing dependence of the AMOC and is thus more
appropriate to scale the strength of the northward flow. Based on these results, in
the following we focus on the level of no motion only. Furthermore, the pycnocline
scale cannot provide any detailed information about the relationship between
overturning and vertical velocity shear of the meridional velocity, which is needed
to understand the wind forcing dependence of the AMOC as we learn later on. Even a
difference of one grid layer likely makes a significant difference in the
accumulation of vertical shear as we have identified above.
\newline

Examining psi_g (ie fig 5) seemed more a distraction than anything fundamental. Thinking about it, the
result in fig 5 must be as such (for dynamical balance), but what is the relation of this analysis to the
main question?

We explain it now in an explicit way.  line 268-279

**\subsection{Maximum overturning and its depth}**
We now analyze the wind forcing dependence of the northward flowing branch of the mid-depth cell. We compute the total maximum overturning streamfunction $\psi_\mathrm{t}$ and the geostrophic maximum overturning streamfunction $\psi_\mathrm{g}$. Conceptually, the differences between $\psi_\mathrm{t}$ and $\psi_\mathrm{g}$ provide insight on the degree to which the depth scale(s) are proxies for the strength of the AMOC. Computing the geostrophic maximum overturning streamfunction $\psi_\mathrm{g}$, the level of no motion is unchanged, but the clockwise (upper) and counterclockwise (lower) rotating overturning cells are substantially altered. The maximum streamfunction $\psi_\mathrm{t}$ includes the surface Ekman flux and the maximum streamfunction $\psi_\mathrm{g}$ exludes the surface Ekman flux. However, the surface Ekman fluxes have to be compensated by an interior return flow that changes in relationship between overturning and its depth. With this section we simply answer the question whether the depth scale does scale overturning over the full depth range of the upper AMOC branch, including the surface Ekman layer, or whether the depth scale does scale overturning below the surface Ekman layer. It is important to answer this question because it does not only provide insight whether the depth scale is a proxy for northward flow but it points also in the direction why the level of no motion does scale northward overturning and why there is a certain relationship between overturning and its depth.
\newline

Similarly I did not really understand fig 6 other than reinforcing fig 4's main result (?)  ITS

It is a technical summary and therefore important for the subsequent analysis which is introduced. We added some more explanatory statements.  line 305-310

Combining our findings from Fig. 4 and 5, we describe the relationship between northward overturning and its depth from a more nonlocal perspective on hemispheric differences. To highlight hemispheric differences in the inter-hemispheric region, we show the meridional averages (30S-10S) and (10N-30N) (Fig. 6). It is technical because we also show the vertical model grid in order to provide an indication for the importance of a single layer change only as well as ability of the depth scales to account for the details in the accumulation of vertical shear. It summarizes the relationship between overturning and its depth. The latter is preparatory for the subsequent analysis. ...

And, not sure what relevant I learned from fig 7 (maybe one could argue the main fig 2 response could be explained as "compensation" effectiveness shown in fig 7?).

We added an explanatory statement.  line 363-372

Fig. 7  shows the zonal-mean meridional velocities ($\frac{\partial {\psi}}{\partial {z}}$) in the wind experiments and the difference in $\frac{\partial {\psi}}{\partial {z}}$ between these wind experiments. It shows the importance for the Ekman cells for the meridional flow and that these Ekman cells cancel out above the level of no motion. Considering the relationship between maximum overturning and the level of no motion, we can think of internal flow in which the Ekman cells are canceled as subsequent analysis reveals. This, in turn, explains the wind forcing dependence of the AMOC. The differences in $\frac{\partial {\psi}}{\partial {z}}$ between the wind experiments are strongest near the equator at the upper levels where the vertical velocity shear changes drastically. Taking the difference

between the 2X and 1X experiments, we find an increase in $\frac{\partial {\psi}}{\partial {z}}$ south of the equator and a decrease north of the equator. To a substantial extent, these changes can be attributed to the strengthening of the local Ekman cells. The differences in $\frac{\partial {\psi}}{\partial {z}}$ at the upper levels between the different experiments demonstrate that the Ekman return flow is baroclinic and occurs mostly above $\eta_\mathrm{\psi}$. The strong influence of the Ekman cells near the surface suggests that, at these levels, the external wind-driven flow associated with the Ekman cells superposes the internal flow that is associated with the level of no motion. These considerations support the perspective that $\eta_\mathrm{\psi}$ is a proxy for $\psi_\mathrm{t}$ rather than a proxy for $\psi_\mathrm{g}$. Small differences in $\psi_\mathrm{t}$ emerge in case of weak compensation of the surface Ekman flux below $\eta_\mathrm{\psi}$.
\newline

 In figure 2, I cannot discern much difference between b and c. Would suggest lighter color shading and black contours at -10,-5,0,5,10,15 etc. It seems to me that the difference in the NH between b and c is very critical to the paper (as shown in fig 4, but not so carefully articulated) but is washed out in the figure here ???

We changed the figure accordingly.

Fig 3, line 212, caption, and other places: to me, plotting density stratification would be drho/dz. What is show here is anomalies in density (normalized by rho_o), plotted in the yz plane (ie y-axis is depth). I would suggest being clearer and more precise in descriptions as such (this is a general issue, and contributes to difficulty in reading). ineWhy are two different g' shown? Scaling relies on the specific estimate not sure g not substantial

We do not change the manuscript because we believe the term stratification as used in our study is clearly defined. Defining a word like stratification is up to us. We show different g' s to illustrate the possible parameter space.

Line 225, "deep isopycnals…" – what does this mean? Depth range
Done
- Lines 249-250 unclear and vague sentence
Done
- Line 252 "deepens … in the northern hemisphere" – doesn't seem to be true 30N-40N (?) downwelling region

We believe it is self-explanatory because we defined the different regions in order to analyze the wind forcing dependence.

- Fig 5 caption: remove "with respect to" (sounds like you are subtracting, or examining anomalies)
Done
- Line 267 Not sure of the purpose of this sentence; I think you mean by definition/construction, the level of no motion and psi_t max coincide? Or am I completely missing something? Suggest removing this sentence and explaining the second sentence better. (see also lines 307-308, 327-328)
Done
- Line 299 "not shown"; wasn't this shown earlier in the paper?
We just clarified.
- Lines 315-317 if you end up keeping the psi_g results, I would provide this explanation earlier on in the paper.

- Line 330 please be more specific as to "mid-depth", didn't see this in plot ok
Done
- Fig 8: I think it might be useful to show mathematically what you compute. Several times it is said "hold the vertical shear constant" – does that mean compute the vertical shear from the 1x experiment (as f(y,z)) but then integrate this quantity using the diagnosed level of no motion in 2xSH and 2x? (lines 348-350). But then why is there a dash line for the 1x case? And, might it be useful to plot the shear, to get some sense as to this quantity between runs?
Done
- Fig 8 caption, line 351: not clear what is meant by "deep". Are you discussing below the level of no motion, or above?
Done
- It was unclear whether the authors felt the (finding of) baroclinic adjustment of the Ekman compensation is a major result. But it would seem to be relevant to the main story, so I think needs to be better woven into the explanation.
Done
We believe with the introduction it is self-explanatory now
- Whether having an eddying model here is critical in these results is also not clear; there are several vague allusions to this, but no clear statement on this point
We added a proper section on the robustness of the results. see below
- I was not entirely satisfied with the justification of the experimental procedure, ie time mean years 10-30 being sufficiently adjusted (I would not have guessed this to be the case, a priori). Perhaps Luschow ref in line 134 could be written more definitively. It would not be inconceivable to have done one high-res run out longer than 30 years to see how well yrs 10-30 adequately captures the quantitative results. Could one assess this using a coarse-res model, or might that muddy any conclusion too much? In any event, it comes across a bit as "take our word for it" rather than having actually attempted due diligence that yrs 10-30 indeed is acceptable for this purpose.

We added a proper section on the robustness of the results.
line 407-456

\section{Robustness of the wind forcing dependence}
In this section we would like to elaborate on the robustness of the results considering the wind forcing dependence of the AMOC. The question arises whether the wind forcing dependence of the AMOC found in the short-term integrations of TP6ML80 (1980-2010) is robust.  In the study we use the time window (1991-2010) in order to allow for major adjustments at an initial stage. We state that the wind forcing dependence found in the time window would reflect a quasi-steady response. This is a strong assumption given that it is actually a transient response within a short integration time. The adjustment in the density field (Fig. 3) support the perspective that major adjustments in ocean dynamics to forcing are realized. However, the wind forcing dependence of the AMOC may still be time-dependent, and low-resolution model outcome may differ from high-resolution model outcome.
\newline

We first show the full time series (1980-2010) of maximum overturning and the level of no motion in TP6ML80 (30S-10S,10N-30N) (Fig. 10 a,b,c,d). There is a strong adjustment and time-dependence in both variables at an initial stage on a decadal time scale. During the course of the study we have neglected this initial adjustment by focusing on the time window 1991-2010 only. After the initial adjustment on a decadal timescale (1980-1990), the wind forcing dependence of maximum overturning and the level of no motion is robust. Nevertheless, there are

oscillations at low frequency which put into question whether the wind forcing dependence of the AMOC found in the short-term integration of TP6ML80 is quasi-steady. We cannot investigate the steady response of the AMOC in TP6ML80 due to the high computational costs. The temporal changes in the level of no motion, however, coincide with the temporal changes in maximum overturning in the sense that the vertical velocity shear of the meridional velocity stays approximately constant over time.
\newline

\begin{figure*}
\includegraphics[width=0.9\textwidth]{time_image}
\caption{The TP6ML80 time evolution (30S-10S,10N-30N) of (a,b) maximum overturning and (c,d) the level of no motion after the forcing is switched on. The MPIESM1.2-LR time evolution (30S-10S,10N-30N)of (e,f) maximum overturning and (g,h) the level of no motion after the forcing is switched on.}
\label{fig:41}
\end{figure*}

\begin{figure*}
\includegraphics[width=0.9\textwidth]{warming_image}
\caption{The time evolution (30S-10S,10N-30N) of (a,c) maximum overturning and (b,d) the level of no motion after the forcing is switched on in the global warming experiments with altered surface wind stress using MPIES1.2-LR. Atmospheric CO$_2$ is quadrupled.}
\label{fig:41}
\end{figure*}

As a next step, we use a AGCM-OGCM coupled low-resolution model to simulate the wind forcing dependence in a low-resolution counterpart and on a longer timescale (50 yr). The coupled model better simulates the salinity balance in the OGCM to which ocean dynamics are sensitive. The coupled model is MPIESM1.2-LR, with the low-resolution configuration of MPIOM being the OGCM component. The ocean model (GR15L40) has a horizontal resolution of 1.5 degrees and 40 vertical levels only. We have a set of four experiments: the 2X experiment in which the zonal and meridional surface wind stress is doubled throughout the hemispheres; the 2XSH experiment in which the wind stress is doubled over the Southern Ocean only; the 1X experiment which is forced under no changes; and the 0.5X experiment in which the zonal and meridional wind stress is halved. We only change the ocean wind stress factor that multiplies the surface wind stress in the coupled model because I am interested in the OGCM dynamics only. It is an online multiplication of each wind stress value at each timestep.
\newline

We find that in the 50 years integrations of the low-resolution model the response is apparently quasi-steady on this timescale (Fig. 10 e,f,g,h). On longer timescales, internal, low-frequent variability may take place. We find that the wind forcing dependence of maximum overturning is similar to TP6ML80 and robust. However, there are major deviations in the level of no motion which does not reflect the wind forcing dependence in the high-resolution model outcome. The general finding that the level of no motion deepens with stronger wind forcing is confirmed, but the details between the 1X and 2XSH experiments are not well simulated. This may be due to model drift in the coupled model, or low-frequency oscillations, or the low vertical model resolution. The level of no motion is sensitive to small variations in the velocity field which may still adjust and oscillate. It seems that the nonlocal wind forcing dependence of the AMOC is less strong and the local wind forcing dependence is much stronger. The vertical velocity shear of the meridional velocity is not constant.
\newline

Disappointed from the finding that the level of no motion may not be well represented in the low-resolution MPIOM configuration, we looked for an alternative way to make sure that the wind forcing dependence of the AMOC is robust. We computed the wind forcing dependence of the AMOC in 100-year global warming experiments with altered surface wind stress, using also MPIESM1.2-LR. We quadrupled atmospheric CO$_2$ and applied the wind stress factor during the forward integration. We initialized with the control experiments with altered surface wind stress at year 30, after having explored that the initialization plays a minor role for the evolution of the AMOC in the global warming experiments. We believe the system is more strongly forced so that the forced underlying dynamics overcome internal oscillations and model drift. Fig. 11 shows the wind forcing dependence (30S-10S,10N-30N) of maximum overturning and the level of no motion in the global warming experiments with altered surface wind stress. Now the wind forcing dependence of maximum overturning and the level of no motion is the same as in the wind sensitivity experiments with TP6ML80.

---

## Author Response (AR3)

**Response Letter**

REVIEW

Nonlocal and local wind forcing dependence of the Atlantic meridional overturning circulation and its depth scale

Minor revision by reviewer 2; accepted by reviewer 1
This response letter addresses the comments by reviewer 2

COMMENT
REPLY

Dear reviewers, dear editor, first of all we would like to thank you for your effort. We were happy to address the minor comments. Thank you that you believe that the paper is publishable. We were disappointed that the second draft was not accepted but we believe that we have made a valuable publication with the final version.

I appreciate the effort made to improve the introduction and motivation (and discussion section); the science here is sound and this paper is a nice contribution to the field.

Thank you for your comment. We believe it does not make sense to list all improvements as it is minor revision with many small improvements at various occasions. Instead, we created a file that gives the difference between the latest draft and the final manuscript which is submitted with this report. Please review this file to get an overview of the changes made.

However my remaining issue is mainly difficulties in understanding due to writing style. Some of my comments below are repeated from previous round of reviews; I would strongly encourage the author take another pass at some improvement, particularly in regard to discussion of Fig 6 and Fig 8, which I think are fairly important to the story. These plots are unfamiliar and need to be carefully explained. As general advice, I think it is better to first explain what is plotted, then go into details about the calculations, rather than finally explain what's plotted buried several sentences after discussing details.

`We changed the sections as well as the overall writing style accordingly.`

Also, section 4 `(5)` is new and apparently in response to comments from both reviewers. I personally was satisfied without this additional analysis (i.e. author's changes made regarding Luschow reference are sufficient), but would not remove the new material. However, I don't think this new material adds to the main result and should be in an appendix or supplementary material section. And, although I did suggest the low-res run, my thought was an idealized efficient setup (ie not coupled) that could be easily run out for 1000-3000 years (or at least a few hundred). I'm not sure the (100-yr) low-res runs do more than muddy the water here, again arguing why this new material is really more appropriate for a supplemental section.

`After having explored different possibilities we believe it is good to have the section on the low-resolution model runs within the main body of the manuscript. It provides an indication for the robustness of the results. We present a new way to understand the AMOC and excite subsequent discussions.`

We addressed the list of the minor comments below. To get an overview of the changes made, please review the file which gives the difference between the latest draft and the final manuscript which is submitted with this response letter. We did further improvements on the manuscript which go beyond the comments of the reviewer.

l. 35 form -> from. But I would argue the sentence "In this way, …" is not necessary at all.

l. 42 the Cessi ref seems a contrary result – but is not mentioned again (in the discussion) to possibly explain her results

l.50 "is substantial" is subjective and vague, suggest being more specific

l. 205, l. 332 "later on" -- be explicit as to which section in the paper you refer

l. 307 "show the meridional averages…" of what? Readers will not immediately grasp what you are showing in Fig 6, explain up front

l. 307 "It is technical" what do you mean by this?

Fig 6 caption: first sentence: "level of no motion (as plotted on the y-axis), in the…" And rewrite the "We do not show" sentence as "Qualitative behavior if plotted instead using eta_rho (not shown) is similar except …"

l. 310 "The latter is preparatory…" delete sentence, do not know what this means

l. 370 "which can be identified in Fig 7" - ???

l 374-394 difficult to follow. Again I'd start off with explaining what is plotted in fig 8, then explain details in calculations, then the science of the results. Complicating the science analyses here, in some sentences I was not sure if the author was comparing 2XSH vs. 2X or 2X{SH] vs. 1x. E.g., In l. 388 are you explaining the difference between the red and red-dashed in Fig 8a? Again the imprecise, indirect writing style makes comprehension problematic.

l. 383 dotted -> dashed

l. 380 and numerous places "deep levels" or "deeper levels" (deeper than what?) Since the paper discusses processes at the surface, in the upper part of the water column, at depth but above the level of no motion, and below the level of no motion, it is not clear what "deep" means. There is no harm in being more specific; if you mean below the level of no motion, say so.

l. 403 "from a generic point of view" - ???

Fig 8 and 9 captions – "velocity shear held constant" this is a bit misleading description. What you

have done is, in your diagnostic, taken the shear from the 1X expt and used this value in computing the diagnostic for transport in the 2X experiment (right?). "Held constant" has a different connotation/expectation of what was done. As someone who has worked on this general problem, I realized immediately what the author was doing here, even appreciated it as clever thing to do. However for a more general reader, the explanation is murky. Never is it clearly posed "does the shear adjust in conjunction with changes in eta_psi (or eta_rho)? To address this, in our diagnostic we employ the shear …"

FIg 8 caption "the time-mean vertical velocity" shouldn't this connect to previous sentence with a semi-colon?

l. 510 puming -> pumping

l. 511 "It forces the flow thus horizontal upstream…" - ???

l. 514 suggest making the point in next sentence without invoking global warming (esp. given no refs provided; maybe link to the section 4 material?) and moreover, it is difficult to say much about temporal adjustment given the 30-yr run period. Or make it clear you are speculating a bit with this motivation? I would keep the point made in the next sentence however.